# HydroBlocks v0.2: Enabling a field-scale two-way coupling between the land surface and river networks in Earth system models

Nathaniel W. Chaney[1], Laura Torres-Rojas[1], Noemi Vergopolan[2], Colby K. Fisher[3]

[1]Department of Civil and Environmental Engineering, Duke University, Durham, NC, USA
[2]Department of Civil and Environmental Engineering, Princeton University, Princeton, NJ, USA
[3]Princeton Climate Analytics, Princeton University, Princeton, NJ, USA

Correspondence to: Nathaniel W. Chaney (nathaniel.chaney@duke.edu)

**Abstract.** Over the past decade, there has been appreciable progress towards modeling the water, energy, and carbon cycles
at field-scales (10–100 m) over continental to global extents in Earth system models (ESMs). One such approach, named
HydroBlocks, accomplishes this task while maintaining computational efficiency via Hydrologic Response Units (HRUs),
more commonly known as "tiles" in ESMs. In HydroBlocks, these HRUs are learned via a hierarchical clustering approach
from available global high-resolution environmental data. However, until now, there has yet to be a river routing approach that
is able to leverage HydroBlocks' approach to modeling field-scale heterogeneity; bridging this gap will make it possible to
more formally include riparian zone dynamics, irrigation from surface water, and interactive floodplains in the model. This
paper introduces a novel dynamic river routing scheme in HydroBlocks that is intertwined with the modeled field-scale land
surface heterogeneity. Each macroscale polygon (a generalization of the concept of macroscale grid cell) is assigned its own
fine-scale river network that is derived from very high resolution (~30 m) DEMs; the inlet/outlet reaches of a domain's
macroscale polygons are then linked to assemble a full domain's river network. The river dynamics are solved at the reach-
level via the Kinematic wave assumption of the Saint-Venant equations. Finally, a two-way coupling between each HRU and
its corresponding fine-scale river reaches is established. To implement and test the novel approach, a 1.0-degree bounding box
surrounding the Atmospheric Radiation and Measurement (ARM) Southern Great Plains (SGP) site in Northern Oklahoma
(United States) is used. The results show: 1) the implementation of the two-way coupling between the land surface and the
river network leads to appreciable differences in the simulated spatial heterogeneity of the surface energy balance; 2) a limited
number of HRUs (~300 per 0.25-degree cell) are required to approximate the fully distributed simulation adequately; 3) the
surface energy balance partitioning is sensitive to the river routing model parameters. The resulting routing scheme provides
an effective and efficient path forward to enable a two-way coupling between the high-resolution river networks and state-of-
the-art tiling schemes in ESMs.

# 1. Introduction

Recent years have seen a renewed effort to improve the representation of land surface heterogeneity in Earth System Models (ESMs) (Chaney et al. 2018; Newman et al. 2014; Martyn P. Clark et al. 2015; Fan et al. 2018). This effort is driven, in part, by the limitations of existing upscaling parameterizations to adequately simulate the multi-scale interactions between the water, energy, and carbon cycles (Wood et al. 2011; Bierkens et al. 2014). Although the desired goal might eventually be a modeling approach that models every meter-scale grid cell over the globe, for the foreseeable future, the significant computational barriers will continue to limit the feasibility of this approach. This is especially true given the need for large ensemble frameworks to handle the significant structural and parameter uncertainties that emerge in models when moving to very high spatial resolutions over large spatial extents (Beven et al. 2015). As a result, there is a persistent need to use reduced-order modeling approaches that are able to converge on the desired meter-scale modeling while minimizing computational expense.

One approach widely adopted over the past decades to represent the landscape heterogeneity within Earth System models (ESMs) is the use of "tiling" schemes, which effectively partition a macroscale grid cell into representative clusters (e.g., forests, grasslands, etc.). This concept is similar to the concept of hydrologic response units (HRUs) in hydrologic models. Since their origin in the late 1980s and early 1990s (Avissar and Pielke 1989; Koster and Suarez 1992), tiling schemes have moved to include spatial variability in land use, soil type, elevation gradients, and management practices, among others. Chaney et al., 2016 and Newman et al., 2014 take this a step further by designing a mechanism to derive these tiles or HRUs by clustering available spatial environmental datasets of the drivers of landscape heterogeneity. Furthermore, over the past decade, there has been an effort to adapt these tiling schemes to enable hillslope-scale processes (e.g., lateral flow) via Darcy flow along topographic gradients within a grid cell (Martyn P. Clark et al. 2015; Fan et al. 2018; Chaney et al. 2018; Subin et al. 2014; Swenson et al. 2019). These additions are making it possible for ESMs to simulate the role of local lateral flow along topographic gradients, which has implications for modeling the riparian ecosystems and surface energy balance partitioning. However, the two-way interconnectivity between the modeled hillslopes and their respective channels has yet to be explored.

In parallel to the development of state-of-the-art tiling schemes, there has been significant innovation in river routing schemes within ESMs in recent years. For a comprehensive review of the advances in macroscale routing schemes the reader is referred to (Shaad and Di Baldassarre 2018); a brief overview of these advances is provided here. Early developments in these schemes focused on the horizontal transfer time and integration of runoff over the land surface, allowing for the validation of early Land Surface Model (LSM) outputs against streamflow observations. This initial work is embodied in the Impulse Response Function Unit Hydrograph (IRF-UH) approach described in (Lohmann, Nolte-Holube, and Raschke 1996), which represented the process as linear and time-invariant, both in grid cells and between grid cells in the river network. Alternative approaches were proposed, focusing instead on storage-based schemes to represent the spatial distribution of flow. The Total Runoff Integration Pathways (TRIP) model presented by (Oki and Sud 1998) was the first to use a simplified form of the 1D Kinematic wave routing equation with a constant velocity for water movement between grid cells, an approach that has since

been implemented in many LSMs over varying scales of global river networks (Pappenberger et al. 2010). With recent computational and data advances, these basic models have been expanded to better utilize the outputs of general ESMs for global predictions beyond just discharge. For example, the recent MizuRoute model (Mizukami et al. 2016) builds upon the approaches described above. MizuRoute combines both approaches to model the flow from hillslope to channels through a gamma-distribution based IRF-UH, while the main channel routing is performed using either a kinematic wave approximation or a linear IRF-UH method, enabling the use of higher resolution terrain and network data with relatively coarse ESM outputs. Alternative approaches for using high-resolution vector river networks have been proposed, such as the Routing Application for Parallel Computation of Discharge model (RAPID). RAPID leverages advances in high-performance computing to create a highly efficient vector river routing scheme that uses a matrix-based version of the Muskingum-Cunge method (David et al. 2016; 2011).

Recent macroscale routing model development has sought to make use of advances in data and computational resources to further enable integrations with ESMs. The Model for Scale Adaptive River Transport (MOSART; (Li et al. 2013)), a recent example of this, divides a grid cell into hillslopes, tributaries, and a main channel to solve the flow of water through the river network via the Kinematic wave approximation of the St. Venant equations. This decoupling of the ESM grid from that of the routing model (network), allows for differences in scale to more accurately capture flow and channel dynamics at any point within a given watershed. Other models have attempted to move beyond the more simplistic representations of flow hydrodynamics to better represent flood inundation. A recent example of this is the Catchment-based Macro-scale Floodplain model (CaMa-Flood) (Yamazaki et al. 2011), which discretizes the grid cells into unit catchments and explicitly parameterizes the topography of the floodplain for each unit. CaMA-Flood then simulates open channel flow via the local inertial equation, a simplified form of the full 1-D Saint Venant equation, allowing for an estimation of water depth and area of inundation in ESM grid cells.

Although there have been significant advances in river routing and tiling schemes over the past decade, there has yet to be a concerted effort to couple these two concepts in ESMs. This persistent lack of interconnectivity between the modeled river network and the land surface leads to: 1) oversimplified sub-grid river networks (e.g., tributaries are mostly ignored) (Jones and Schmidt 2017; Swanson and Meyer 2014; Rice 2017); 2) a lack of interaction between the land surface tiles and the river networks (e.g., simulated flooding of the Nile River doesn't recharge the land surface) (Shen et al. 2016; Helton et al. 2012; Bisht et al. 2017); 3) the water moving through the sub-grid river network does not influence the surface energy partitioning (i.e., macroscale schemes continue to mostly treat river networks as pipes) (Zampieri et al. 2011; Bisht et al. 2017; Sheng et al. 2017); 4) local-scale irrigation and water management schemes are spatially agnostic and rely almost exclusively on the main channel (Shaad and Di Baldassarre 2018; Voisin et al. 2017; Pokhrel et al. 2015). All these deficiencies compound to illustrate a persistent weakness in Earth system models by ignoring critical processes that are known to play an important role in both the natural and engineered hydrologic systems (Pokhrel et al. 2016; Fan et al. 2018). With the recent advances in both river routing and tiling schemes, the timing is right to enable two-way interactions between the tiling schemes and river networks in ESMs.

This study provides a path to address this persistent deficiency in ESMs. This is accomplished by implementing a reach-based routing model in the HydroBlocks land surface model (Chaney, Metcalfe, and Wood 2016) and enabling a two-way coupling with the modeled hydrologic response units. The primary features of the novel river routing scheme include: 1) each macroscale polygon (a generalization of the concept of macroscale grid cell) is assigned its own field-scale river network delineated from DEMs; 2) the fine-scale inlet/outlet reaches of the macroscale polygons are linked to assemble the continental river networks (and to ensure conservation of mass); 3) river dynamics are solved at the reach-level via an implicit solution of the Kinematic wave simplification of the St. Venant equations; 4) a two-way coupling is established between HRUs and the river network. The scheme is implemented over a 1.0-degree bounding box around the Southern Great Plains site in Northern Oklahoma in the United States. Furthermore, a series of experiments are performed to understand: 1) the sensitivity of the land surface to the two-way coupling; 2) the number of HRUs that are required to approximate the fully distributed simulation adequately; 3) the impact of the uncertainty in the routing scheme parameters in the macroscale response.

## 2. Data and Methods

### 2.1 Study Domain

A 1.0-degree box in central northern Oklahoma and southern Kansas in the United States is used to implement and test the new routing scheme (see Figure 1). The region is generally flat with a shallow decreasing gradient in terrain, precipitation, and NDVI from west to east. The overall climate is typically dry, with cold winters and hot summers. The Salt Fork Arkansas River, Chikaskia, and to a lesser extent the Arkansas River traverse the region; all rivers in the domain eventually flow into the Arkansas River. The vegetation throughout the domain is primarily croplands with forested regions found along the riparian areas. There are also small urban areas dispersed throughout the region with the town of Enid in the southeast corner being the largest. This domain contains the Atmospheric Radiation and Measurement (ARM) Southern Great Plains central facility (SGP CF) among other ARM SGP facilities. As the largest climate facility in the world, ARM SGP collects a wealth of data on land-atmosphere interactions and atmospheric processes that are used frequently to evaluate and improve sub-grid atmospheric processes in ESMs. Understanding the role that the interconnected land surface and river network play in the partitioning of the surface energy balance (and its role in land-atmosphere interactions) is a primary motivation for using this domain in this study.

### 2.2 Land surface model: HydroBlocks

HydroBlocks is a field-scale resolving land surface model (Chaney et al., 2016) that accounts for the water, energy, and carbon balance to solve land surface processes at high spatial and temporal resolutions. HydroBlocks leverages the repeating patterns that exist over the landscape (i.e., the spatial organization) by clustering areas of assumed similar hydrologic behavior into HRUs. The simulation of these HRUs and their spatial interactions allows the modeling of hydrological, geophysical, and biophysical processes at the field scale (e.g., 30 m) over regional to continental extents (Chaney, Metcalfe, and Wood 2016;

Vergopolan et al. 2020). The core of HydroBlocks is the Noah-MP vertical land surface scheme (Niu et al. 2011). HydroBlocks applies Noah-MP in an HRU framework to explicitly represent the spatial heterogeneity of surface processes down to field scale. At each time step, the land surface scheme updates the hydrological states at each HRU; and the HRUs dynamically

interact laterally via subsurface flow. In the original HydroBlocks, subsurface flow between HRUs was modeled via a subsurface kinematic wave. In this implementation, following the approach used in Chaney et al. (2018), the subsurface flow module has been updated by computing the Darcy flux between adjacent HRUs at each subsurface level. The fluxes are then included as divergence terms within their corresponding subsurface level of the vertical one-dimensional solution of Richards' equation in Noah-MP. This allows for the flow of water between HRUs to also be driven by capillarity and not just the

predefined topographic gradient.

## 2.3 Hierarchical generation of hydrologic response units

The HRU generation scheme in the original HydroBlocks (Chaney et al., 2016) was not sufficient to capture the dynamics in riparian zones (e.g., runoff), which led to the development of the hierarchical multivariate clustering (HMC) approach introduced in (Chaney et al. 2018). This study builds on the 2018 version of HMC to enable adequate coupling of the land

surface with the river network within HydroBlocks. HMC uses a field-scale map (e.g., 30 meters) of delineated watersheds and the corresponding DEM as building blocks to generate the HRUs. Prior to constructing the HRUs, following Chaney et al. (2018), the river network and watersheds are delineated from a 30-meter DEM by first sink-filling and then delineating the channels using an area threshold of 10,000 m$^2$. The 30-meter channel pixels are then grouped by reach. The watersheds are assembled by finding all 30-meter pixels that flow into a given reach via steepest descent. The following steps are then taken

to assemble the HRUs from these data.

1) *Macroscale polygons* – The large domain is partitioned into smaller subdomains defined in this study as macroscale polygons. A macroscale polygon is a generalization of the concept of a macroscale grid cell; it is analogous to a grouped response unit (GRU; M. P. Clark et al. 2015). HMC is designed to handle any geometry of macroscale polygons; the only constraint is that a given delineated watershed cannot be split between polygons. This constraint precludes the use of a

regular latitude/longitude macroscale grid. Instead, for this study, a 0.25 arc degree grid is modified to meet this constraint. To be more specific, the boundaries of each 0.25 arc degree grid cell are adjusted such that a watershed is assigned to the macroscale polygon where it has the majority of its area (see Figure 2A for an example). Note that other domain decompositions are feasible under HMC's current architecture.

2) *Clusters of watersheds* – The watersheds contained within a given macroscale polygon are clustered using K-means. The

feature predictors used in the clustering include latitude, longitude, flow accumulation area, and the natural logarithm of the flow accumulations area. Log-scale accumulation area was used to separate the low-order reaches from the high-order reaches, accumulation area was used to separate reaches within the large rivers, and latitude/longitude were used to represent location-driven differences in atmosphere-forcing, land-use, and soils, among others. Figure 2B shows an example of this division for 10 clusters of watersheds (*k*) of the macroscale polygons. After clustering the watersheds, to

maximize similarity among the watersheds that are assigned to a given cluster of watersheds, the empirical cumulative distribution function (CDF) of the 30-meter height above nearest drainage (HAND; (Nobre et al. 2011)) values within each watershed is matched to a representative watershed's HAND empirical CDF. The representative watershed's HAND empirical CDF of a cluster of watersheds is set to be the average of all of its watersheds' empirical CDFs (for a background on CDF matching, see (Reichle and Koster 2004)). Note that the concept of a cluster of watersheds is only used for the

non-routing component of HydroBlocks.

3) *Height bands* – The 30-meter HAND values of each cluster of watersheds are discretized into height bands. First, all 30-meter pixels that belong to the channels within a given cluster of watersheds are grouped into one height band. Then, the non-channel 30-meter pixels are discretized into additional height bands by binning HAND values. The binning involves assembling the smallest HAND values that have an areal coverage $n$ (user-defined) times larger than its adjacent lower

height band. For example, the height band immediately above the channel height band will have an areal coverage $n$ times larger than the channel height band. The next height band will have an areal coverage $n^2$ larger than that of the channel and so forth. The described method is a simple yet robust approach to maximize spatial detail of height bands near riparian zones while coarsening upslope. Figure 2C shows the discretization into height bands of the watersheds that belong to the cluster of watersheds 1 of the macroscale polygon 7 in the domain. An additional *maxhb* parameter is used to ensure the

number of height bands per cluster of watersheds does not exceed this user-defined threshold (*maxhb* is fixed at 100 in this study).

4) *Intra-band clusters* – To characterize the role of sub-height band heterogeneity of land use, soils, and elevation, among others on water and energy states and fluxes, each height band within each cluster of watersheds is further divided into clusters (here called intra-band clusters). This is accomplished by iterating per height band and clustering all the

corresponding 30-meter pixels. Given that the areal coverage of each height band can vary; the number of intra-band clusters per height band is set to be proportional to their fractional coverage of the cluster of watersheds. The user-defined $p$ parameter is the average number of clusters per height band. Figure 2D shows an example of this division for $p = 5$ using latitude, longitude, land cover, and clay as covariate features. The fractional coverage weight of $p$ compensates for the telescoping mesh of height bands produced by the parameter $n$. The smaller height bands next to the

channel will have few intra-band clusters while the larger ones upslope will have many more.

The multi-level hierarchy of clusters of watersheds, height bands, and intra-band clusters leads to the definition of the hydrologic response units (HRUs) per macroscale polygon (Figure 2E). In summary, each macroscale polygon is divided into $k$ clusters of watersheds. A given cluster of watersheds groups together several reaches (one reach per watershed). Note

that this clustering of watersheds is only used in the non-routing components of HydroBlocks; each reach maintains its unique properties in the routing scheme. All watersheds that belong to a given cluster of watersheds are split first into height bands (defined by $n$ and *maxhb*) and then intra-band clusters ($p$). The coupling between the land surface and routing modules handles the interaction between clusters of watersheds and the simulated reaches.

## 2.4 River routing scheme

The river routing scheme implemented within HydroBlocks is a reach-based implicit solution of the St. Venant equations with the Kinematic wave assumption. The conservation of mass across a 1-dimensional reach is given by:

$$\frac{\partial A}{\partial t} = \frac{\partial Q}{\partial x} + S$$

Where $\frac{\partial A}{\partial t}$ is the change in the cross-sectional area of the flow with respect to time $\left[\frac{L^2}{T}\right]$, $\frac{\partial Q}{\partial x}$ is the change in discharge across an infinitesimally small reach length (note that $Q = uA$ where $u$ is the velocity of the flow), and $S$ is a source/sink term of the

200 cross-sectional area. Following the Kinematic wave assumption, uniform flow is assumed (i.e., $s_0 = s_f$), and thus $u$ can be estimated at each time step for a given reach via Manning's equation:

$$u = \frac{1}{n} R_h^{\frac{2}{3}} S_0^{\frac{1}{2}}$$

Where $n$ is Manning's coefficient, $R_h$ is the hydraulic radius, and $S_0$ is the slope of a given reach. Note that the hydraulic radius is given by $R_h = \frac{A}{P}$ where $P$ is the wetted perimeter. For this implementation, a compound channel composed of a

205 rectangular channel and a symmetric floodplain is assumed (see Figure 3). The segmented conveyance method is used where the conveyance of both channel ($c$) and floodplain ($f$) components are summed, and the effective velocity for the compound channel is computed as follows:

$$u = \frac{S_0^{\frac{1}{2}}}{A}\left(\frac{1}{n_c} A_c R_{h,c}^{\frac{2}{3}} + \frac{1}{n_f} A_f R_{h,f}^{\frac{2}{3}}\right)$$

As shown in Figure 3, each reach's cross-sectional profile is composed of a rectangular channel and a symmetric floodplain

that is learned from the reach's discretized height bands (computed in Section 2.3). At each reach, the cross-sectional area of the flow and the derived cross-sectional profile are used to determine the wetted perimeters $P_c$ and $P_f$ and the cross-sectional areas $A_c$ and $A_f$ at each time step.

To solve the St. Venant equation with the Kinematic wave assumption at each time step, a fully implicit first-order finite volume upwind scheme in space and backward Euler in time is used:

$$\frac{A_i^{n+1} - A_i^n}{\Delta t} = \frac{Q_{i+1}^{n+1} - Q_i^{n+1}}{\Delta x} + S$$

where the upstream $Q_{i+1}^{n+1}$ contribution to a given reach is the sum of the discharge of all the reaches that are immediately upstream of a given reach ($Q_{i+1}^{n+1} = \sum_j u_j^{n+1} A_j^{n+1}$). In other words, the upstream internal boundary of reach $i$ is the sum of the discharge of all reaches $j$ that flow into reach $i$ (Jacovkis and Tabak 1996). Note that $\Delta x$ is the length of a given reach in the network and will rarely be the same across reaches. $S$ is a source/sink term that can account for inflow from runoff as well as water abstractions from irrigation and recharge of the land surface. This leads to a system of non-linear equations (per macroscale polygon) to be solved per time step. Given the non-linearities in the hydraulic radius calculation as well as the need to iteratively exchange network boundary conditions between macroscale polygons, an iterative time step is implemented. In the current implementation, a Picard iteration is used to attempt convergence at each time step.

Similar to (Mizukami et al. 2016), an impulse response function is used to route the runoff produced at the HRU level in HydroBlocks to its corresponding channel. HydroBlocks' HRU delineation is first used to assemble a histogram of travel times (constant velocity of the flow per grid cell) of all 30-meter pixels that belong to a given HRU to their closest channel. The travel times consider a constant velocity of the flow per grid cell (fixed to 0.1 m/s in this study). Note that although the constant velocity parameter is set to 0.1 m/s, it is an HRU-specific parameter that can be modified by the user. The flowpath across the landscape is computed via d8 flow direction. At each time step, the convolution of this histogram (~unit hydrograph) with the HRU runoff is used to compute the runoff from a given time step that reaches the channel at future time steps.

## 2.5 Assembling the stream network and river reach cross-sectional profiles

For each macroscale polygon, the river network is derived at a 30-meter spatial resolution following the method described in Chaney et al. (2018). For each river reach, the channel width and bankfull depth are computed using the functional relationships derived for the Contiguous United States described in (Bieger et al. 2015). To minimize inconsistencies between reaches that belong to the same cluster of watersheds, the computed channel width and bankfull depths are averaged across all corresponding reaches. The cross-sectional profile for each reach (not cluster of watersheds) is learned by first extracting all the 30-meter pixels in a given watershed; the profile follows the height band discretization from Section 2.3. The precomputed channel length is used to compute each section of the profile's width, given the corresponding height band's area within that reach. Because the spatial resolution of the DEM (30 meters) is much larger than many of the computed channel widths of the delineated streams (~1 meter), the additional fractions of the channels' collocated 30-meter pixels that are not part of the channel are assigned to the height band right above the channel (i.e., the lowest element of the floodplain). Figure 3 shows an example of the computed profile for reach 122 in macroscale polygon 7. To avoid "step changes", the shown linear interpolation between the hand values of each height band is used when computing the cross-sectional area $A_f$ and wetted perimeter $P_f$ of a given channel's floodplain (as shown in Figure 3). A different cross-sectional profile is made per reach in a given macroscale polygon. It is important to note that the routing scheme is run on the complete DEM-derived sub-grid network of river reaches and not the clusters of watersheds (defined in Section 2.3).

## 2.6 Two-way interaction: Coupling of the land surface (HRUs) and the river reaches

This section explains how the HRUs interact with the routing scheme's river reaches. After the river routing scheme (i.e., Kinematic wave) is updated for a given time step (Figure 4A), each reach's derived cross-sectional profile (see Section 2.5) is used to determine the inundation height over each of the height bands of that reach. This is done by effectively pouring the volume of water contained within the reach and calculating the inundation heights that correspond to a flat surface at the top of the river. To assemble the inundation heights for a given cluster of watersheds, the inundation heights per height band are averaged across all watersheds that belong to a given cluster of watersheds (Figure 4B); these computed inundation heights are then equally distributed to their corresponding HRUs (Figure 4C). The computed inundation height per HRU is then added to the following time step's update of Noah-MP as a constant flux at the soil surface (Figure 4D). Water is not only able to infiltrate the soil in the HRU via Noah-MP, but it is also able to move laterally to another HRU via HydroBlocks' modeling of subsurface flow (see Section 2.2 for more details). The total column of runoff produced during the time step at the given HRU is set as the HRU's new inundation height. The difference between the old and new inundation heights are computed (Figure 4E) and averaged up to the height band level (Figure 4F). To determine the changes of inundation height at each reach (not cluster of watersheds), the computed differences in inundation heights at the cluster of watersheds level are scaled to the reach level. This is accomplished by multiplying the change in inundation height value by the ratio between the watershed's original inundation height per height band and its corresponding cluster of watersheds inundation height. These differences are then aggregated into a difference in cross-sectional area $\Delta A$ per watershed (Figure 4G). Finally, this difference is divided by the model time step and added to the reach's source/sink term $S$ in the routing scheme (Figure 4H). Note that the source/sink term also includes runoff originated from the land surface that is not inundated; this water arrives at the reach via the method described in Section 2.4. After updating the cross-sectional area per reach via the river routing scheme, the two-way coupling begins again for the next time step (Figure 4A).

## 2.7 Model experiments

A series of model experiments are run over the study's domain to evaluate the implementation of the two-way coupling between the land surface and the routing model within HydroBlocks. For all experiments, the model is run between 2015 and 2017 at an hourly time step. Furthermore, to parameterize and force HydroBlocks', a suite of high-resolution datasets are used, including the 1 arcsec (~30 meter) USGS National elevation dataset (Gesch et al. 2009), the 1 arcsec (~30 meter) POLARIS soil properties database (Chaney et al. 2019), the 1/32 degree (~3 km) Princeton CONUS Forcing (PCF) dataset (Pan et al. 2016) that provides meteorological forcing at 1 hour temporal resolution, and the 1 arcsec (~30 meter) National Land Dover Database (Fry et al. 2011).

Boundary conditions are provided at the domain inlets of the Salt Fork Arkansas and Chikaksia rivers. Boundary conditions for the Arkansas river are not provided since it only covers a very small fraction of the domain in the northeast quadrant. For the Salt Fork Arkansas River, the discharge data from the Great Salt Plains Lake reservoir is used; and for the

Chikaksia River, the USGS station at Corbin, Kansas, is used. The domain inlets do not exactly match-up with the location of the gauges as the observed gauges are a few km upstream. However, the benefits of including these data as boundary conditions appears to outweigh not using them.

For all experiments, the covariates used for identifying the clusters of watersheds are latitude, longitude, flow accumulation, and the natural logarithm of the flow accumulation area. The covariates used for the intra-band clusters (i.e. HRUs) are the 30-m resolution latitude, longitude, land cover, and percent clay. An overview of the set of experiments run are outlined below.

Exploratory simulation – A baseline simulation is run to provide an initial overview of the features of the new model. The parameters of the introduced HRU generation scheme (see Section 2.3) are set to be k = 10, $n$ = 2, and p = 5.

Sensitivity to the two-way coupling – To evaluate the impact of the two-way interaction between the modelled rivers and the land surface, two model simulations are performed. For both simulations, the HRU generation scheme is the same as that used for the baseline experiment. The first one uses the default approach of not allowing the routing scheme to interact with the land surface, while the second enables the two-way connectivity scheme introduced in Section 2.6.

Convergence – Similar to the approach used in Chaney et al. (2018), nine different model experiments are run to evaluate how the HRU configuration parameters impact the model simulations. Simulations A-C focus on inter-watershed heterogeneity by increasing $k$ from 1 to 5 to 10, while setting $p$ = 1 and $n$ = 1000. Simulations D-F focus on the discretization of the height bands by decreasing $n$ from 5 to 3 to 2, while setting $p$ = 1 and $k$ = 10. Finally, simulations G-I focus on the role of intra-band heterogeneity by increasing the average number of intra-band clusters from 2 to 3 to 5, while setting $n$ = 2 and $p$ = 5.

Sensitivity to routing model parameters – To determine the importance of the routing model parameters in Hydroblocks, following the approach used in Chaney et al. 2016, a Sobol sensitivity analysis is performed. The Sobol sensitivity analysis (Sobol 1993) is a method that decomposes the variance of the model output into contributions from each parameter (first-order index $S_i$) and its interactions with other parameters (total-effect index $S_{Ti}$). The parameter ensemble used in the sensitivity analysis is assembled by sampling two ensembles of size $n_s$ from the Sobol quasi-random sequence (Sobol 1993) and then cross-sampled by holding one parameter fixed for a total of $n_s(k_s + 2)$ parameter sets, where $k_s$ is the number of parameters. In this study, $n_s = 64$ and $k_s = 5$ leading to a total of 448 parameter sets. Each parameter set is used to run a separate HydroBlocks simulations leading to 448 different ensemble members.

The modeled time series of domain-average latent heat flux, sensible heat flux, land surface temperature, and inundation height are then compared to the baseline coupled simulation via the root-mean squared error (RMSE). These calculated RMSE values are then used to compute the first-order sensitivity indices ($S_i$) and the total-effect sensitivity indices ($S_{Ti}$). The model parameters used in the sensitivity analysis include the Mannings roughness coefficients for the channel and

floodplains ($n_c$, and $n_f$), the channel width ($w$) the bankfull depth ($b$), and the uniform overland flow velocity ($v$). Given that each reach is assigned its own values of for $n_c$, $n_f$, $w$, and $b$ when assembling the model over the domain, to minimize complexity, a set of scalar multiplier parameters are used to scale the precomputed parameters of all reaches uniformly. The range for all the scalar multipliers is set to be 0.25–4. This is not the case for the uniform overland flow velocity ($v$) which is sampled in logarithmic space between 0.1-1 m/s and assigned equally to each reach.

## 3. Results

### 3.1 Exploratory simulation

As an initial baseline simulation, the updated HydroBlocks model is run between 2015 and 2017 at an hourly resolution over the study domain using the HRU generation scheme parameters defined in Section 2.7. This section focuses on a simulated inundation event to provide a general understanding of the simulated land surface, routing scheme, and floodplain dynamics. Figure 5 shows the simulated inundated height at the peak of the flooding event on August 11th, 2017 at 14:00 UTC. Note that only the central four 0.25-degree grid cells in the domain are plotted for visual clarity. The time series of simulated discharge at four of the reaches in the area are shown as well. The inundation results show how flooding occurs primarily in the watersheds in the northwest and southeast grid cells. In all cases, the flooding appears to be a flash flood event as the main river that traverses this section (Salt Fork Arkansas River) only barely floods. Closer inspection of the time series of reach 14 in macroscale polygon 6 and reach 8 in macroscale polygon 7 (both track the Salt Fork River) shows the role that the tributaries play in increasing the flow after reach 14 but before reach 8. The latter secondary flood wave at both reaches most likely indicates regional influences provided by the boundary conditions into the domain upstream. Since the discharge of the Salt Fork River at the inlet to the domain originates from a reservoir, this most likely explains the dampened flood response.

To further investigate the impact that this flooding event has on the land surface, Figure 6 shows the mapped simulated inundation height, root zone soil moisture, and latent heat flux over the same four central 0.25-degree grid cells on August 5th 14:00 UTC, August 11th 14:00 UTC, and August 14th 14:00 UTC. The chosen time steps coincide with a dry period before the simulated event, right after the rain event, and a few days after the event. A brief description of the simulated states and fluxes for the three days is provided below.

August 5th, 2017 14:00 UTC - All rivers within the domain are within their banks with relatively low stage height within the channels. Not surprisingly, the root zone soil moisture is very low throughout the entire area, with the notable exceptions being the Salt Fork Arkansas River (west to east) and the Chikaksia River (northeast), and lakes throughout the region (note that, in the current implementation, lakes are modelled separately from the implemented river routing scheme). The tributaries throughout the region also have higher soil moisture due mostly to recharge from redistribution of runoff from the land surface via the river network. A similar story is evident in the mapped latent heat flux, with the only noticeable difference being the higher latent heat fluxes in the south of the domain.

August 11th, 2017 14:00 UTC – This time step falls within the flooding event. Several tributaries of the Salt Fork
Arkansas River are flooding, and the stage height in the Salt Fork Arkansas River is appreciably higher than that only 6 six
days before. The flooding signal is immediately apparent in the root zone soil moisture, where the channel and adjacent HRUs
are close to or at saturation. These differences are not as noticeable in the latent heat flux, most likely since the entire area is
at or close to field capacity due to the widespread rain event and thus the evapotranspiration is not constrained by soil moisture.

August 15th, 2017 14:00 UTC - The floodwaters have receded, and all rivers are again within their banks, although
the stage height of the Salt Fork Arkansas River is still higher when compared to August 8th. Except for the channel HRUs,
root zone soil moisture has decreased when compared to August 11th. However, the imprint of the flood event is still evident
in the riparian zones.

The simulated event in Figures 5 and 6 provide a first-order understanding of how all the pieces of the implemented
land surface and routing scheme come together to directly impact surface fluxes, soil moisture, and flooding over the domain.

**3.2 Sensitivity to the two-way coupling**

Two different model experiments were run to further investigate the effect of the two-way coupling on the modeled states and
fluxes. The first simulation is called "uncoupled", and it uses the default approach (i.e., not allowing the routing scheme to
interact with the land surface). The second simulation is called "coupled", and it enables the connectivity scheme introduced
in Section 2.6.

Figure 7 shows the difference in simulated annual mean sensible heat flux, latent heat flux, root zone soil moisture,
and land surface temperature over the four central macroscale polygons in the domain. For each variable, the results are shown
for both the uncoupled and the coupled simulations. These simulations would suggest that over this region, the local lateral
flow of subsurface flow is only one of the contributors to sustaining the riparian zones (as represented in HydroBlocks). The
redistribution of water via surface flow, flooding, and recharge also plays a key role in maintaining these ecosystems.

Figure 8 investigates the role of this coupling further by plotting the time series of the spatial mean and spatial standard
deviation of sensible heat flux, inundation height, latent heat flux, land surface temperature, and root zone soil moisture. In
general, the results show how the coupling can lead to differences in both the spatial means and spatial standard deviations.
The differences are more appreciable for the spatial standard deviation, as would be expected from the spatial maps from
Figure 7. While the relative change in the spatial mean between the two is on the order of 0.1-2 %, the relative change in the
spatial standard deviation can be around 20-50% for variables such as latent heat flux and sensible heat flux. The differences
are more extreme during the summer when the region is drier and, thus, a recharge mechanism for the riparian zones will play
a much more important role (as seen in Section 3.1). Overall, the results appear to show that adding a two-way coupling
between the land surface and routing scheme will increase the macroscale evaporative fraction (decrease Bowen ratio).

**3.3 Convergence**

A convergence analysis was performed to determine how well the reduced-order model representation in the updated HydroBlocks model is representative of the heterogeneity of the domain. More specifically, the model was run for increasingly complex HRU configurations. Simulations A-C focus on inter-watershed heterogeneity by increasing $k$ from 1 to 5 to 10 while setting $p = 1$ and $n = 1000$. Simulations D-F focus on the discretization of the height bands by decreasing $n$ from 5 to 3 to 2 while setting $p = 1$ and $k = 10$. Finally, simulations G-I focus on the role of intra-band heterogeneity by increasing the average

number of intra-band clusters from 2 to 3 to 5 while setting $n = 2$ and $p = 5$.

Figure 9 illustrates how the increase in heterogeneity complexity impacts the fine-scale simulated sensible heat flux. More specifically, it shows the annual mean sensible heat flux over the domain. The explanation of the results below is split into understanding the role of each parameter:

A. Increasing the number of clusters of watersheds (increasing $k$; Figure 9A-C) – Initially, there are two HRUs represented
per 0.25-degree grid cell; one channel and one "floodplain". In this scenario, upon recharge of the floodplain, the characteristic inundation heights are effectively an area-weighted average between all of the inundation heights of all reaches (remember that the flow through each reach is still resolved in the routing scheme). Increasing the number of clusters of watersheds from 5 to 10 leads to differences in the connectivity between the rivers and the land surface. Increasing the number of clusters of watersheds leads to a separation between main channels and tributaries and, thus, a
distinction between their interaction with their respective riparian zones. Increasing the number of clusters of watersheds also leads to an increase in spatial heterogeneity due to the ability to represent spatial differences in land cover and soil properties. For example, with 10 clusters of watersheds the southwest macroscale polygon is able to start to represent the urban settlements, such as the town of Enid.

B. Increasing the number of height bands (decreasing $n$; Figure 9D-F) – As the $n$ parameter decreases (ratio between the area
of a height band and its adjacent height band below it), the number of height bands increases. This enables a finer discretization at the channel/floodplain interface, which in turn leads to more realistic cross-sectional profiles per reach and thus improved floodplain dynamics. When the number of height bands is too low, the inundation height that should correspond only to the region immediately adjacent to the channel is instead evenly distributed to a much larger area upslope, thus, diffusing its influence on recharge of the riparian area.

C. Increasing the number of intra-band clusters (increasing $p$; Figure 9G-I) – The increase of intra-band clusters leads to a substantial increase in the heterogeneity within each macroscale polygon. The most noticeable difference is the emergence of the urban areas and the country roads and interstates (although these are not clearly visible at 1.0 degree). The other changes are driven by the separation between crops, grasslands, forests, and bare soil as well as soil properties. Note how, in this case, the boundaries between the macroscale polygons effectively disappear.

Figure 10 formalizes the convergence analysis by showing how the temporal mean of the spatial mean and standard deviation of the plotted 30-meter maps per 0.25 degree varies as a function of the number of HRUs. For all cases, the largest

differences in spatial means are across the different macroscale polygons, which illustrates the controlling role of climate and the model land characteristics in the macroscale polygon mean values for the majority of the variables. This can also be seen in Figure 9, where the west to east gradient in precipitation/temperature/vegetation is readily apparent in the modeled sensible heat flux. For each case, the largest differences in the spatial mean occur when increasing the number of clusters of watersheds. For the spatial standard deviation (which can be interpreted as a metric of heterogeneity), the changes are more abrupt, with the largest changes occurring when increasing the number of clusters of watersheds. For all macroscale polygons in the domain, the convergence is relatively quick (although one could argue that root zone soil moisture has not yet converged). It is encouraging that all macroscale polygons follow similar paths and the use of 300-350 HRUs appears to be sufficient to adequately model the fine-scale features while maintaining computational efficiency — the 16 interconnected subdomains (i.e., macroscale polygons) take 5 minutes per year of simulation on 16 cores. This setup coincides with $k = 10$, $n = 2$, and $p = 5$, which is the reasoning behind its use throughout the paper. It should be noted that this convergence analysis is non-exhaustive since it only looks at the predefined HMC parameter path configuration (see Section 2.7). For example, the role of the number of clusters of watersheds might be different if one starts by increasing the number of intra-band clusters instead of the number of clusters of watersheds. Work is ongoing among the co-authors to find the optimal path configuration to minimize the number of HRUs even further.

### 3.4 Parameter sensitivity

To assess the role of routing module parameters' in the two-way coupling, a 448 member Sobol Sensitivity analysis is run. This sensitivity experiment explores the role that the Mannings coefficients (channel and floodplain), the bankfull depth, the channel width, and uniform flow velocity have on the simulated macroscale states and fluxes. These results focus exclusively on 1st July-15th July of 2017 since it covers a time period that shows a rapid wetting of the soil, followed by a prolonged drying period—ideal environmental conditions to explore the role of the two-way coupling.

Figure 11 illustrates the ensemble spread in the simulated spatial mean and spatial standard deviation for a suite of states and fluxes time series; the baseline uncoupled simulation that was used in Section 3.2 is also shown for comparison. The biggest difference between the uncoupled and coupled simulations happens primarily during the dry down period, where the latent heat spatial mean and spatial standard deviation are significantly enhanced with respect to the baseline simulation; the impact of the coupling on the spatial mean of most variables is negligible for almost all parameter sets. The parameter sensitivity plays an important role during the dry periods, with the flooding and recharge components playing a key role in these differences. As expected, the spread in the surface fluxes is most pronounced during the middle of the day since this is when available soil moisture will play the largest role in transpiration. Even though there is a clear signal in the sensitivity of the coupling to the parameters, the general role that the river plays is consistent for most parameter sets.

Figure 12 takes this sensitivity analysis a step further by exploring the role of each parameter. The results show that for sensible heat flux, latent heat flux, root-zone soil moisture, and land surface temperature, the channel width ($w$) and channel bankfull depth ($b$) play the largest role. In the case of the the spatial mean of latent heat flux and sensible heat flux, there are

important interactions of the channel width parameter ($w$) with other parameters. In the case of runoff, the channel mannings coefficient ($n_f$) plays the largest role with important secondary interactions most likely with the channel width ($w$). Finally, the inundation height is almost primarily controlled by the channel mannings coefficient ($n_c$). These results highlight the key role of channel geometry in determining the role of river networks on the macroscale fluxes and states; in practical terms, the role of channel width can be simply explained by the fact that a larger surface area of water will produce a larger latent heat

flux over the domain. Although further analysis is necessary, this result is most likely irrespective of the two-way coupling. In the case of inundation height, the role of the channel Mannings coefficient is most likely simply due to how it defines the stage-height of the channel (and thus inundation height); these results would most likely vary if the inundation height only factored in floodplain water. In all cases, the role of the floodplain mannings coefficient is minimal indicating only a short period of time where there was inundated water on the floodplain. Finally, the role of the uniform overland flow velocity is

small but not negligible for all cases.

## 4. Discussion

### 4.1 Evaluating the two-way coupling parameterization

The primary goal of this work is to develop and implement the two-way coupling parameterization within HydroBlocks. Although the results show the sensitivity of the model to the two-way coupling, it remains unclear if this paramerterization

improves the macroscale modeling of surface fluxes and inundation. Preliminary comparisons (not shown here) between domain-averaged in-situ observations of surface fluxes and land surface temperature show negligible improvements; this is expected from Figure 8 which shows minimal changes in spatial means between the coupled and uncoupled simulations. A more realistic evaluation of the parameterization should involve using observations that focus on the riparian zones and floodplains which is where the impact of these two-way coupling will be most prominent. In the absence of new in-situ

measurements, one approach is to use high resolution remote sensing of land surface states and fluxes. More specifically, for land surface temperature one could use GOES (~ 2 km), MODIS (~1 km), and Ecostress (~ 100 m). Other products that would be useful include field-scale evapotranspiration products derived from high resolution satellite remote sensing data (e.g., Anderson et al. 2011). Beyond surface fluxes, the modeled inundation could be compared to remotely sensed maps of flood extent (e.g., Horritt and Bates 2002) or against a hydrodynamic model (e.g., HEC-RAS; Brunner 2010). All of these data would

provide a more complete picture of the utility of the parameterization. Furthermore, the added-value of this parameterization will most likely be larger in arid river basins such as the Niger and Nile river basins; basin-wide implementations of HydroBlocks over these regions would provide a more robust macroscale evaluation of the parameterization.

## 4.2 Disconnect between clusters of watersheds and reach-based river routing

One of the challenges of the two-way coupling implemented in HydroBlocks is the spatial disconnect between the hydrologic response units and the river reaches. This problem is mostly due to the clustering of watersheds; each of the watersheds within a cluster of watersheds is assumed to behave similarly, even if their river reaches are higher-order or in a different region of a sub-polygon. The choice of cluster predictors is made to ameliorate these challenges. As can be seen in Figure 2B, when assembling the clusters of watersheds, the watersheds are grouped based on their attributes. For watersheds that are along the main channel, using accumulation area as an attribute is able to separate these higher-order reaches from the lower ones; however, the existing coupling scheme still allows for water that is on the floodplain downstream within a given cluster of watersheds to be placed upstream at the beginning of the time step. This unrealistic "diffusive" intra-cluster redistribution is the primary limitation of this approach. For smaller-order streams, the grouping generally disregards the higher-order stream hierarchy. This means that lower-order streams are split between higher-order streams, which can lead to cross-basin redistribution. Future work should look into more appropriate ways to cluster these watersheds. One could imagine only allowing the clustering of watersheds to happen for lower-order reaches; the higher-order reaches could then be left as-is. This would effectively place the computational burden on the main channels of a given macroscale polygon while simplifying the tributaries. One could also only enable the two-way coupling for clusters of watersheds that are sufficiently physically consistent (e.g., main channels).

## 4.3 Implementing dynamic lakes, reservoirs, and surface water abstraction

The reach-based routing scheme explored in this study provides the scaffolding to model dynamic lakes and reservoirs. Assuming that the outflow positions of water bodies along the channel can be assigned to a model reach and that estimates of the bathymetry of the water bodies can be "burned" into the corresponding reach cross-sections, this should be relatively straightforward to implement. The flooding component of the existing routing scheme enables a reach's corresponding valley to fill-up and, thus, produce a first-order representation of the time-varying lake spatial coverage. Given that many lakes will cover multiple reaches, intra-reach water bodies will need to be included. Although an ad-hoc parameterization could be used with the existing kinematic wave routing scheme, future updates should move towards a diffusive wave approach to be able to handle backflows. Finally, given the unavoidable split of water bodies among macroscale polygons, careful attention will need to go to towards inter-macroscale polygon water bodies. However, there is no reason that the existing approach could not handle it; the only increase in computational burden would be to pass more information between macroscale polygons.

One of the primary motivations for implementing a coupling between a routing scheme and the land surface in HydroBlocks is to eventually enable the implementation of surface water management (e.g., irrigation from surface water abstraction and reservoir operations). The proposed implementation of lakes could be readily adapted for reservoirs. The primary difference is that the output flows would be controlled by operation rules instead of solely through lake storage. Furthermore, the reach-based implicit method used in this study would allow for surface water abstraction where each HRU

that has irrigation can draw water from either the most accessible reservoir (as defined by the closest reach that is categorized as a reservoir), or directly from its closest channel.

## 4.4 Improving the modeling of channel and floodplain dynamics

Although the routing scheme implemented in HydroBlocks is able to represent the role of floodplains in the compound channel flow (see Section 2.4), there is still a lack of proper representation of the separation between channel and floodplain flow.

Moving forward, an alternative would be to adapt the extended Saint-Venant equations used in HEC-RAS (Brunner 2010). This approach solves the flow on both the channel and the floodplain by coupling two solutions of the Saint-Venant equations via flow between the channel and floodplain. In addition, the implemented routing scheme uses the kinematic wave approximation of the Saint-Venant equations. However, this assumption is not suitable for many flat regions (e.g., Mississippi, Amazon). The simplest path forward appears to be the implementation of a diffusive assumption of the Saint-Venant equations.

The added value of this approach would extend beyond a more realistic movement of water through the channel. It would also be useful as lakes and reservoirs are included in HydroBlocks, which will require the modelling of backwater effects. To address this problem, the current scheme should be replaced with a higher-order implicit scheme in future work. Finally, future efforts could also drop the uniform flow velocity assumption used to model overland flow along the hillslopes; this could be amended in a future version of HydroBlocks by calculating dynamic flow velocities (e.g., kinematic wave) when modelling

HRU to channel overland flow (similar to MOSART (Li et al. 2013)). However, given the relatively low sensitivity of the model to this parameter (see Figure 12), the added-value of dynamic overland flow velocities is unclear.

## 4.5 Leveraging existing river network databases

In this study, the river network was delineated directly from the NED 30-meter elevation dataset (Gesch et al. 2009). This was done to ensure the consistency between the river network and the DEM which is necessary to form a rigorous coupling between

the HRUs and the river reaches. As this approach is implemented over continental to global extents, this approach should be need be revisited. Over the United States, follow-up work should use of the National Hydrography Dataset (NHD) (USGS 2018).  The NHD is a vector database that defines the spatial locations and connectivity of lakes, ponds, streams, rivers, canals, dams, and stream gages. Most importantly, this database includes the modified NED raster DEM, thus making a robust connection between the vector reaches and associated DEM. These data will also make it possible to consider endorheic basins

as well as existing reservoirs and lakes. Over global extents, the MERIT hydro-vector database provides the clearest path forward (Lin et al. 2021). In any case, to be able to adequately use these data, there will be a need to further upgrade the existing routing scheme. As it currently is implemented, the existing approach is only able to couple a river with a channel width up to the fine-scale pixel length (e.g., 30 meters) to the land surface. Future implementations will amend this assumption to enable coarser rivers to interact with the floodplain.

## 4.6 Optimizing macroscale polygon geometries for the two-way coupling scheme

Preliminary results suggest that the current scheme has load balancing weaknesses on High-Performance Computing (HPC) when implemented over larger domains (e.g., Contiguous United States). This appears to be primarily due to the disparate number of river reaches between macroscale polygons. In other words, the domain decomposition (i.e., the geometries of the macroscale polygons) does not balance the sub-polygon river networks. Given the need for reaches to communicate between macroscale polygons, the most complex sub-grid river network slows down the solver over the entire domain. A promising path forward is to optimize the macroscale polygon geometries to enforce more balanced sub-polygon river network topologies. This is not a new challenge; existing macroscale vector-routing schemes such as RAPID (David et al. 2016) and MizuRoute (Mizukami et al. 2021) have implemented optimal domain decomposition strategies. However, these domain decomposition approaches are not designed for a two-way coupling between HRUs and river reaches. When HRUs and their corresponding river reaches are on different computational cores in HydroBlocks, the computational slowdown due to message passing between computational cores generally exceeds that due to load balancing issues. Furthermore, the number of HRUs per macroscale polygon can vary; this is especially true if the number of HRUs is optimized per polygon. The mismatch between the number of reaches and HRUs, the need for the river reaches and HRUs of a given macroscale polygon to be on the same computational core (or at least same shared memory), and the unbalanced number of river reaches across the domain are the current roadblocks for ESM implementation that must be resolved moving forward.

## 4.7 Reducing sub-polygon complexity: Clustering lower-order river reaches

Beyond optimizing macroscale polygon geometries to improve inter-polygon connectivity, there is a need to more adequately constrain the number of sub-polygon river reaches. Indeed, although the number of reaches per macroscale polygon in this study (~400 reaches per polygon) remains computationally tractable, preliminary results suggest that this is not always the case. This problem will be especially important in ESMs where the number of sub-polygon HRUs/tiles must be kept to under a few dozen; and thus, the routing module would consume a significant portion of the land model's compute time. Furthermore, the increase in the number of reaches for coarser macroscale polygons would quickly add a significant computational burden (e.g., 2,500-20,000 reaches per 1.0-degree grid cell). Therefore, moving forward, there is a need to simplify the sub-polygon networks. One approach being explored by the co-authors is to cluster the lower stream orders that fall within a given macroscale polygon; these river reaches can then be modeled as one. Given that the first-order reaches usually take up more than half of the reaches of a macroscale polygon, if these were clustered and the numerical solver was adapted to handle this framework, this would reduce the number of reaches significantly. This approach could be applied to all tributaries while maintaining the sub-grid reaches of the main channels that traverse the macroscale polygon (similar to the approach used in MOSART (Li et al. 2013)).

## 5. Conclusions

The existing lack of interconnectivity between the modeled river network and the land surface in ESMs leads to: 1) oversimplified macroscale river networks (e.g., tributaries are mostly ignored); 2) no existing mechanism for sub-polygon river networks to interact with the land surface HRUs; 3) water moving through the river network does not influence the surface energy partitioning; 4) sub-polygon irrigation and water management schemes are spatially agnostic and rely mainly on the main channel. This study illustrates a path forward to address these persistent deficiencies in ESMs. This is accomplished by implementing a reach-based routing model in HydroBlocks and enabling a two-way coupling with the modeled hydrologic response units. The primary features of the novel river routing scheme include: 1) each macroscale polygon is assigned its own field-scale river network delineated from DEMs; 2) the fine-scale inlet/outlet reaches of the macroscale polygons are linked to assemble the continental river networks; 3) river dynamics are solved at the reach-level via an implicit solution of the Kinematic wave simplification of the St. Venant equations; 4) a two-way coupling is established between each cell's sub-grid tiles and the river network. The experiments run over the study domain illustrate the sensitivity of the land surface to recharge due to floodplain dynamics and the role that it can have on surface energy partitioning. Furthermore, the appreciable sensitivity of the model to reach characteristics reinforces the need for improved channel morphology parameterizations and datasets.

**Code and Data availability.** The model code and data used in this study are available at: https://doi.org/10.5281/zenodo.4071692 and https://doi.org/10.5281/zenodo.4070128.

**Author contribution.** N. C., N. V., and C. F. conceived the research. N. C. and L. T. developed and implemented the model. N. C. led the writing of the paper and N. V., C. F., and L. T. contributed to the writing.

**Conflicts of interest.** The authors declare that they have no conflict of interest.

**Acknowledgements.** This study was supported by funding from NOAA grant NA19OAR4310241 (Parameterizing the effects of sub-grid land heterogeneity on the atmospheric boundary layer and convection: Implications for surface climate, variability and extremes) and NOAA grant NA19OAR4310244 (3D-Land Energy Exchanges: Harnessing High Resolution Terrestrial Information to Refine Atmosphere-to-Land interactions in Earth System Models).

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

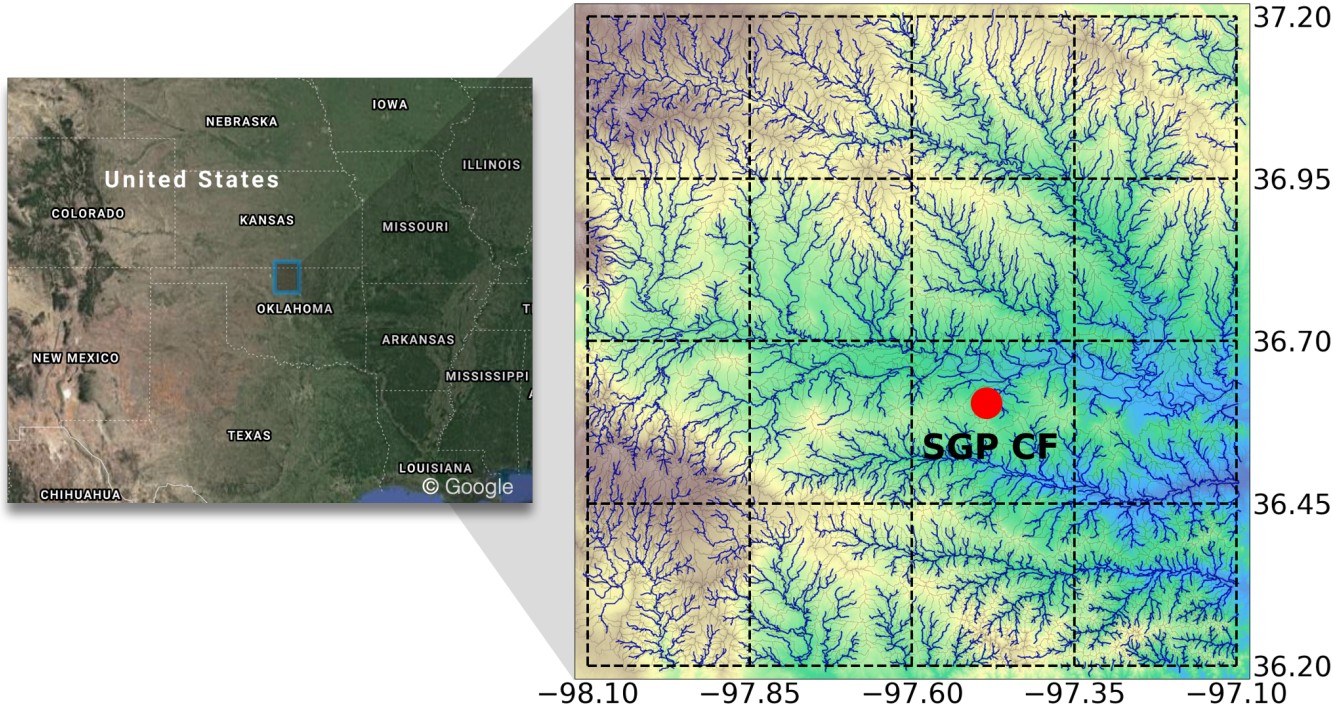

**Figure 1. The study area consists of a 1.0-degree box in central northern Oklahoma and southern Kansas. The domain contains the Atmospheric Radiation and Measurement (ARM) Southern Great Plains central facility (SGP CF).**

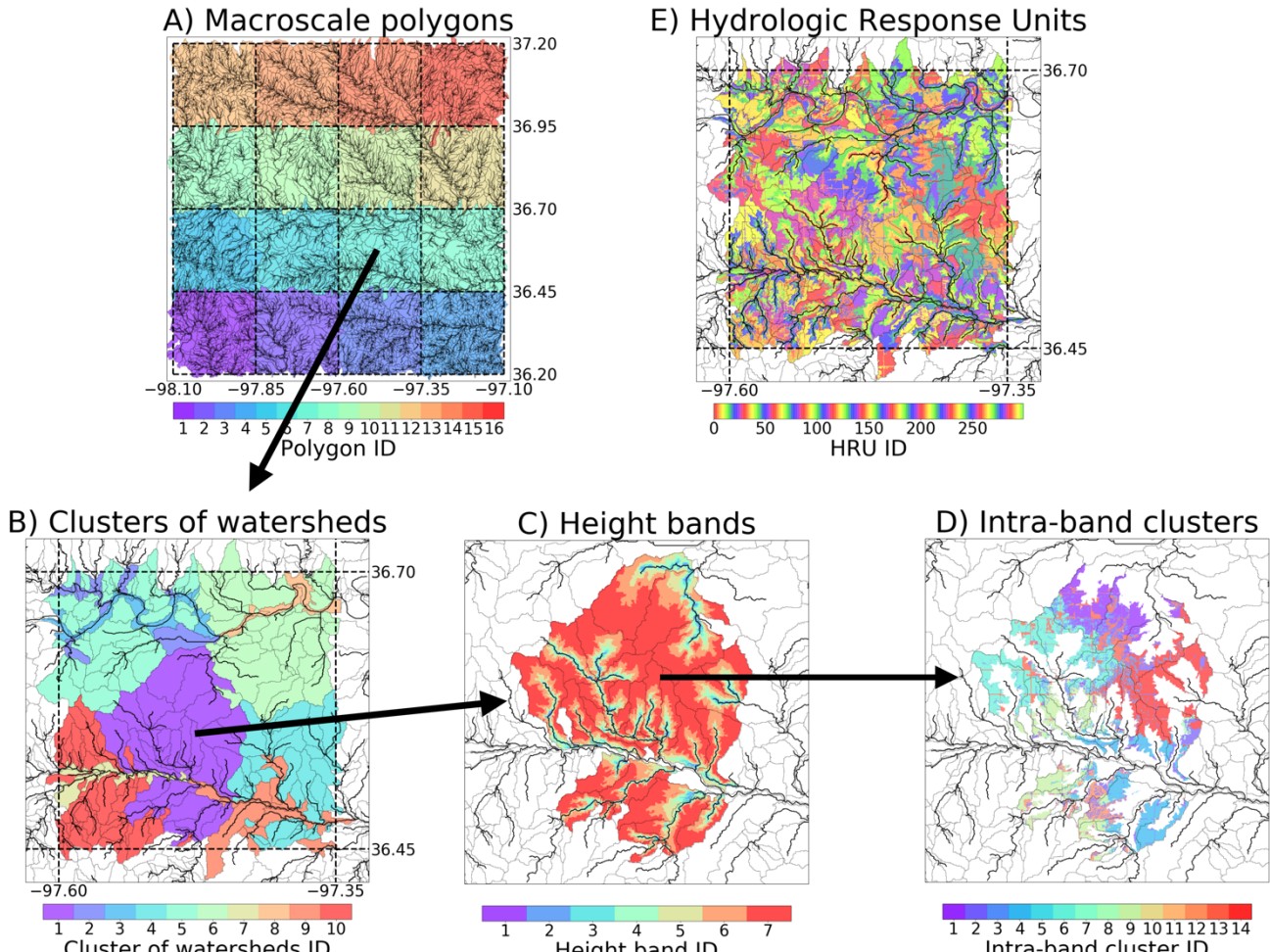

**Figure 2. (A) The domain is split into 16 macroscale polygons, where each color identifies a polygon. (B) Each macroscale polygon is partitioned into 10 clusters of watersheds via K-means clustering. (C) The HAND data is used to discretize each cluster of watersheds from channel to ridge; this discretization allows to assemble the channel and non-channel height bands. Finally, (D) each of the height bands is split into intra-band clusters (D). The C and D methods are applied to all clusters of watersheds in B to assemble the final HRU map (E).**

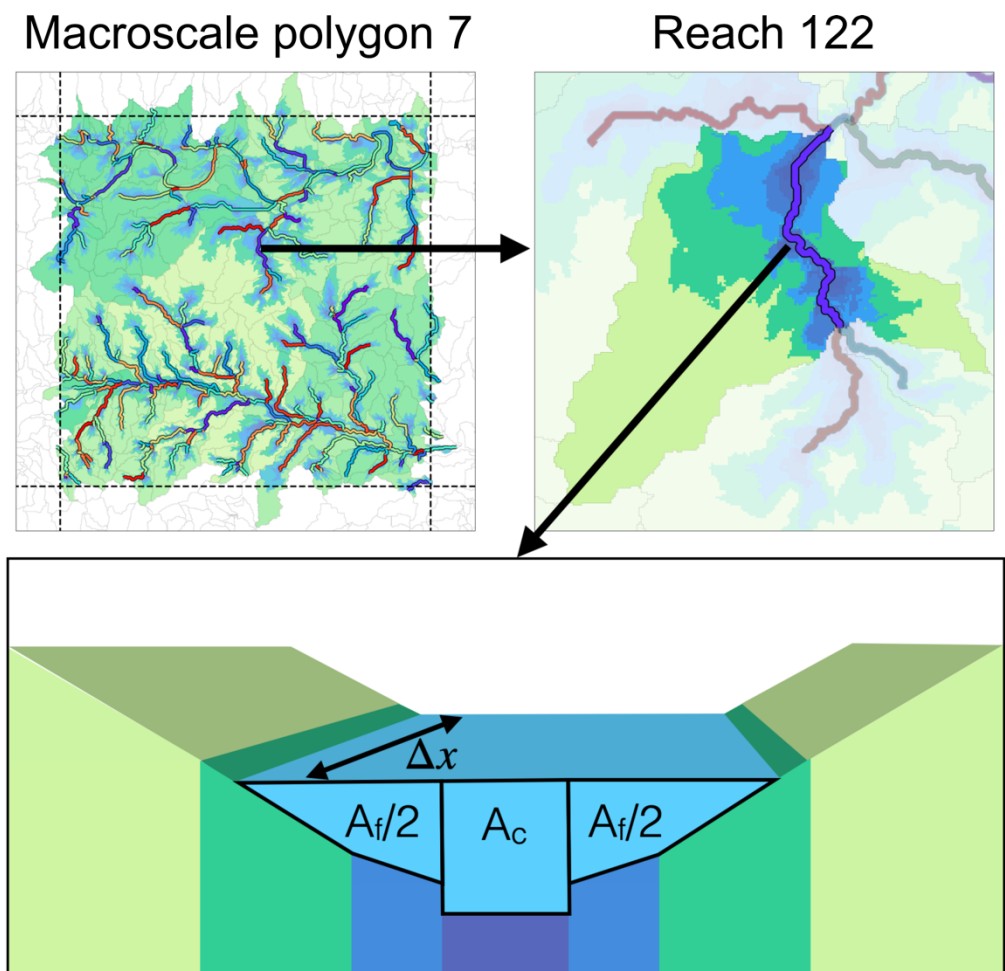

**Figure 3. The illustration of a river reach in the domain. Using the HAND values, the river reaches are split into a channel (dark blue) and floodplain component (ligher blue and greens), where the floodplain component is a combination of the floodplains and hillslopes. Note that a few height bands are omitted in the diagram for simplicity. Similar profiles are assembled for each reach in each macroscale polygon.**

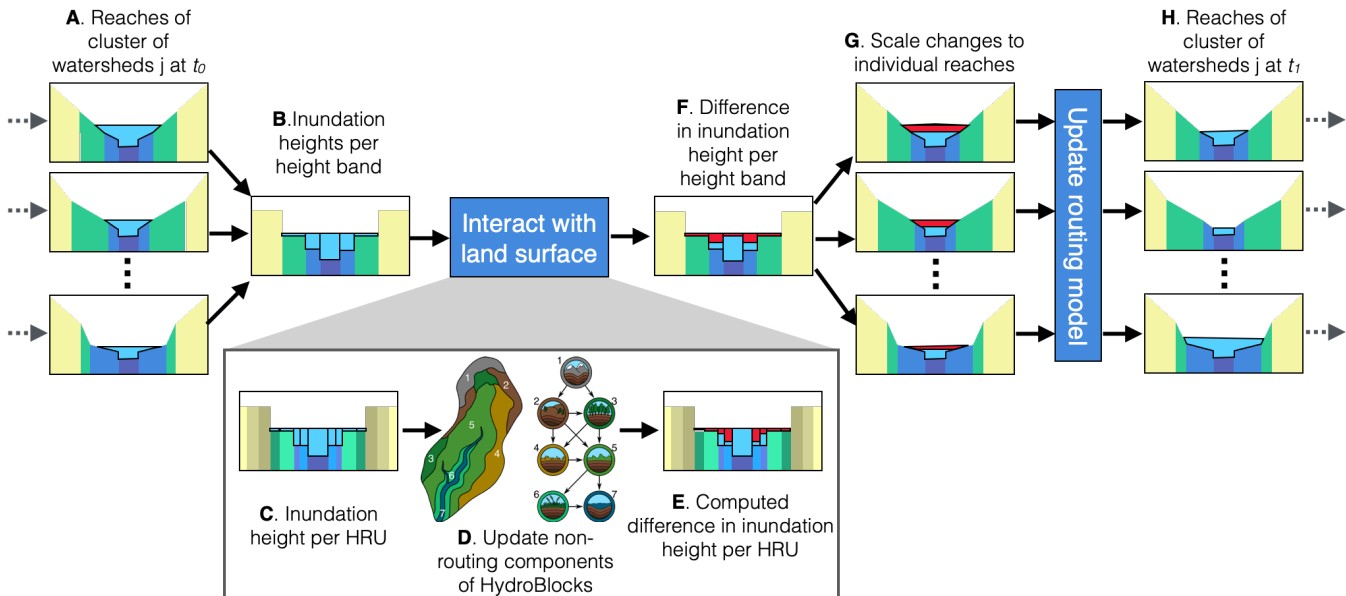

Figure 4. Floodplain inundation scheme implemented in HydroBlocks. After the routing module is updated, for a given time step (A), each reach's computed cross-section profile is used to determine the inundation height over each of the height bands in a given reach. The inundation heights per height band are averaged across all watersheds in a given cluster of watersheds (B). These inundation heights are then equally distributed to their corresponding HRUs (C). To update the land surface model component, the computed inundation heights are added to the following time step's as a constant flux at the soil surface to update the non-routing components of HydroBlocks (D). The difference between the old and new inundation heights are computed (E) and averaged back up to the height band level (F). The differences per height band at the cluster of watersheds level are first scaled to each watershed and then aggregated to compute the change in cross-sectional area (G). This difference is then divided by the routing model time step and added to the routing module's source/sink term $S$ in the next iteration of the routing model (H).

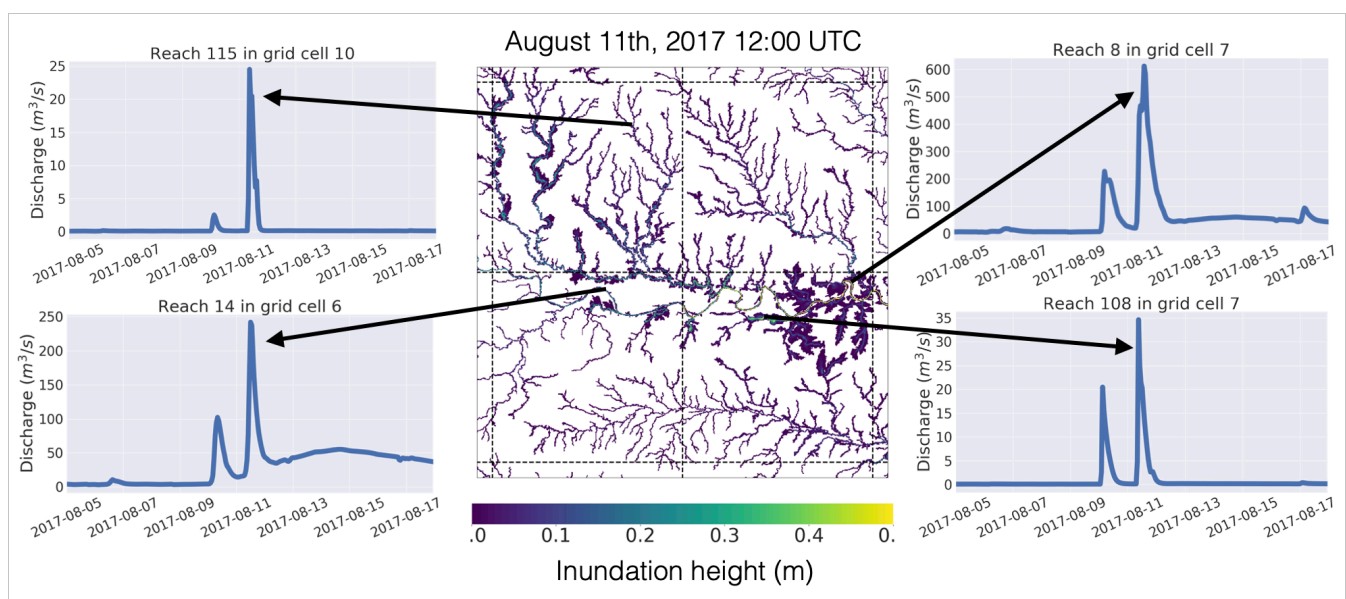

**Figure 5. The center panel shows the simulated inundated height over the four central macroscale polygons in the study domain on August 11th, 2017 at 12:00 UTC. The remaining four plots show the simulated discharge at four different reaches between August 5th and August 17th. The arrows point to the time step on each time series that correspond to the mapped inundation height. Note that mapped inundation height is upscaled to a 100-meter spatial resolution for visual clarity.**

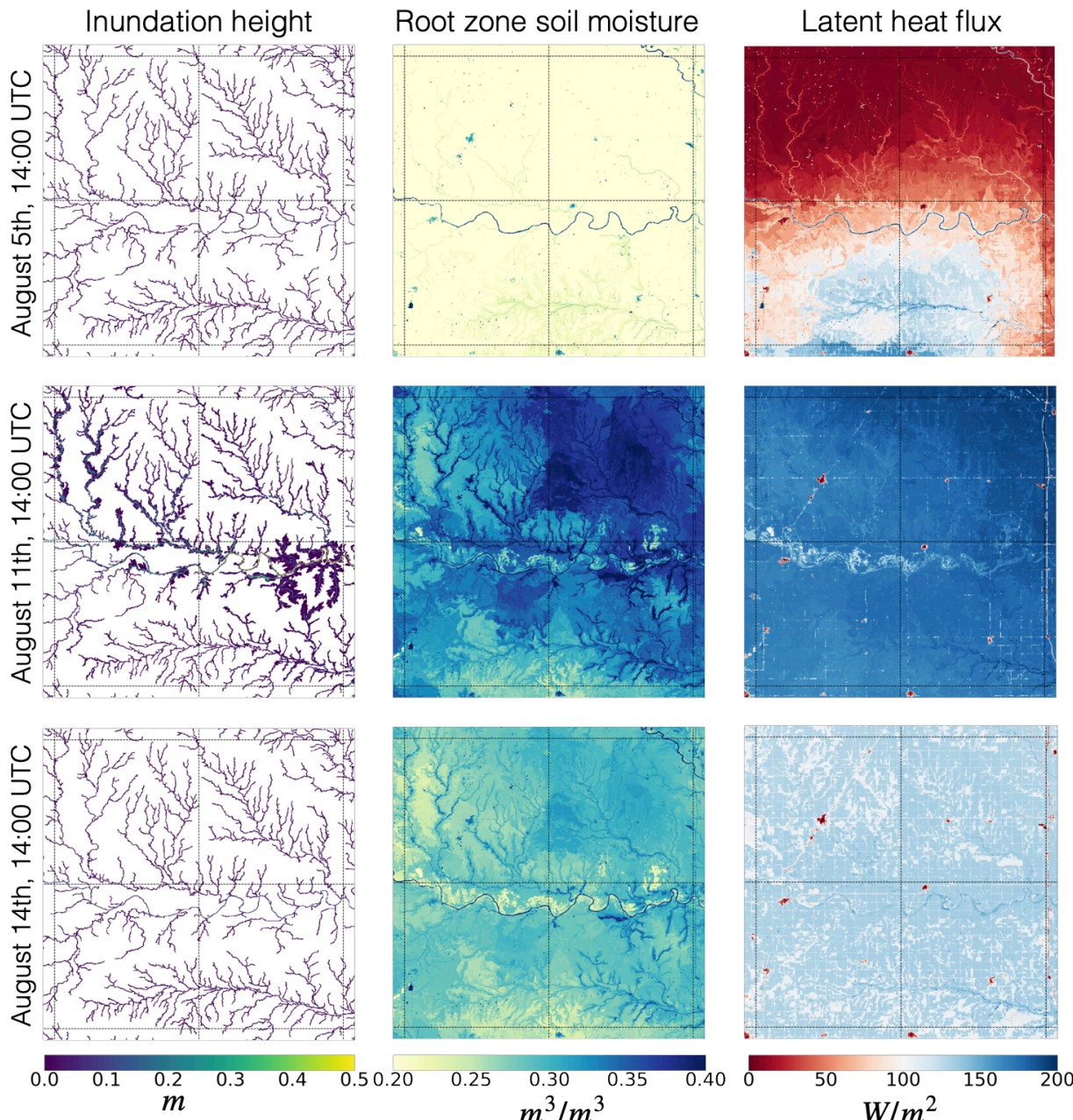

**Figure 6. Mapped simulated inundation height, root zone soil moisture, and latent heat flux over the central four macroscale polygons in the study domain on three different time steps: August 5th 14:00 UTC, August 11th 14:00 UTC, and August 14th 140:00 UTC. The chosen time steps coincide with the event shown in Figure 5 and show the area before, during, and after the flood event.**

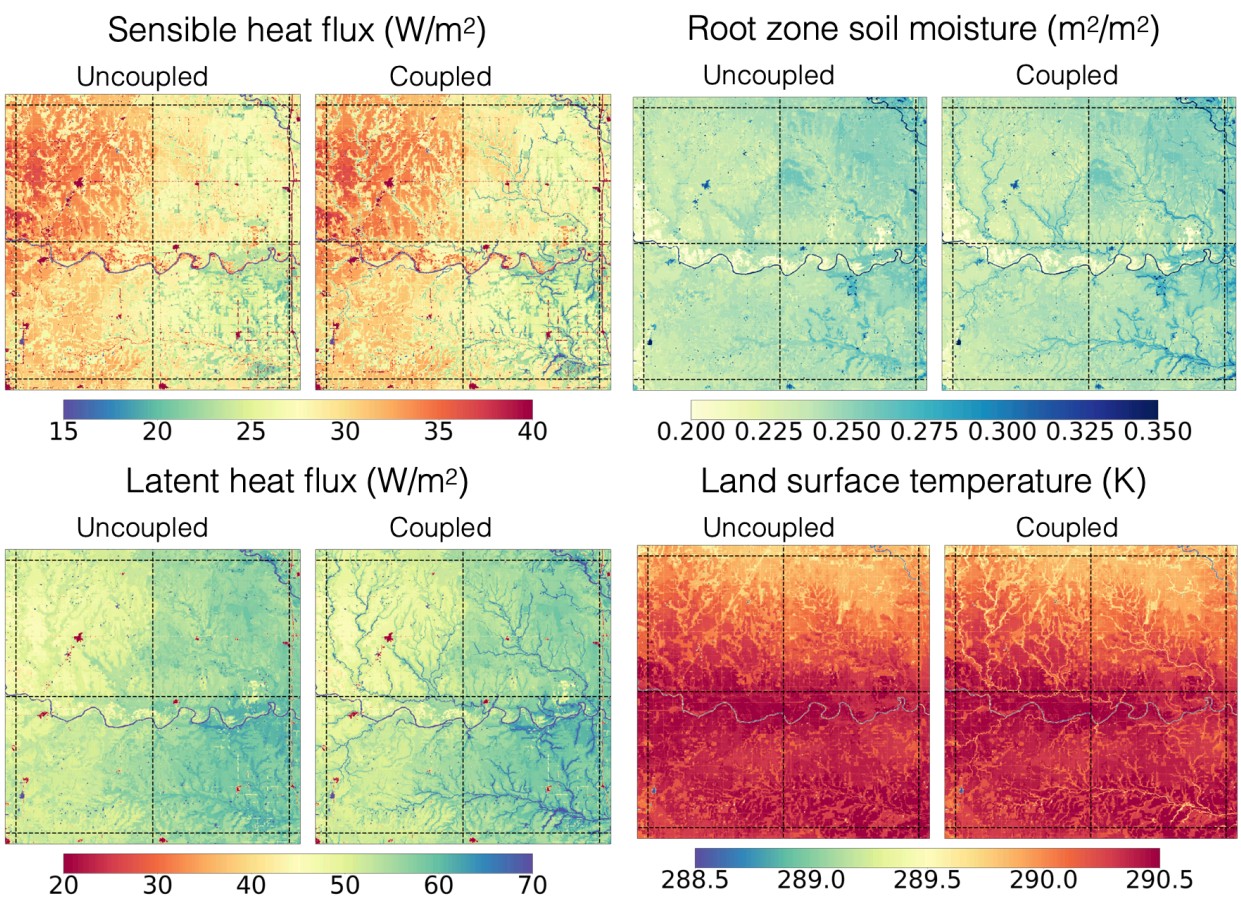

**Figure 7. Annual mean sensible heat flux, latent heat flux, root zone soil moisture, and land surface temperature. The left column shows the results for the uncoupled simulations (i.e., the routing scheme does not interact with the land surface), while the right column shows the coupled simulations. Data was upscaled to a 100-meter resolution for clarity.**

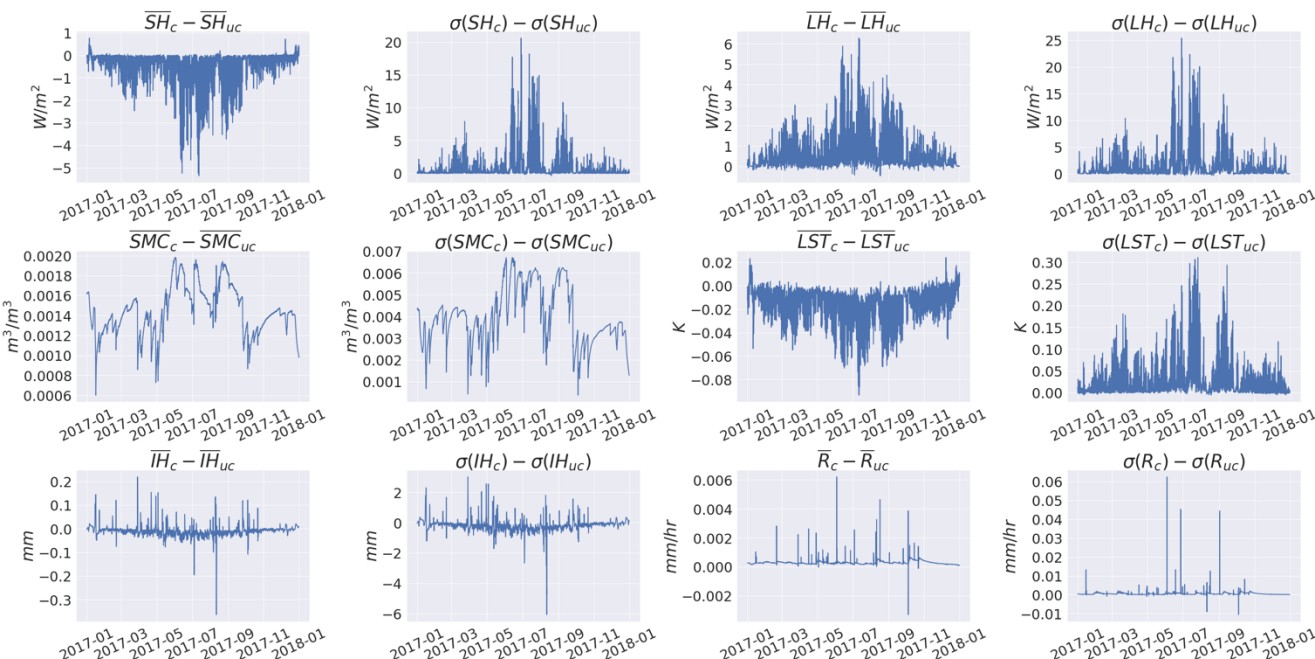

**Figure 8. Temporal differences between the coupled and uncoupled simulations between January 1st, 2017 and December 31st, 2018.** The explored variables include the spatial mean ($\bar{X}$) and spatial standard deviation ($\sigma(X)$) of sensible heat flux (*SH*), latent heat flux (*LH*), root zone soil moisture (SMC), land surface temperature (*LST*), runoff (*R*), and inundation height (*IH*).

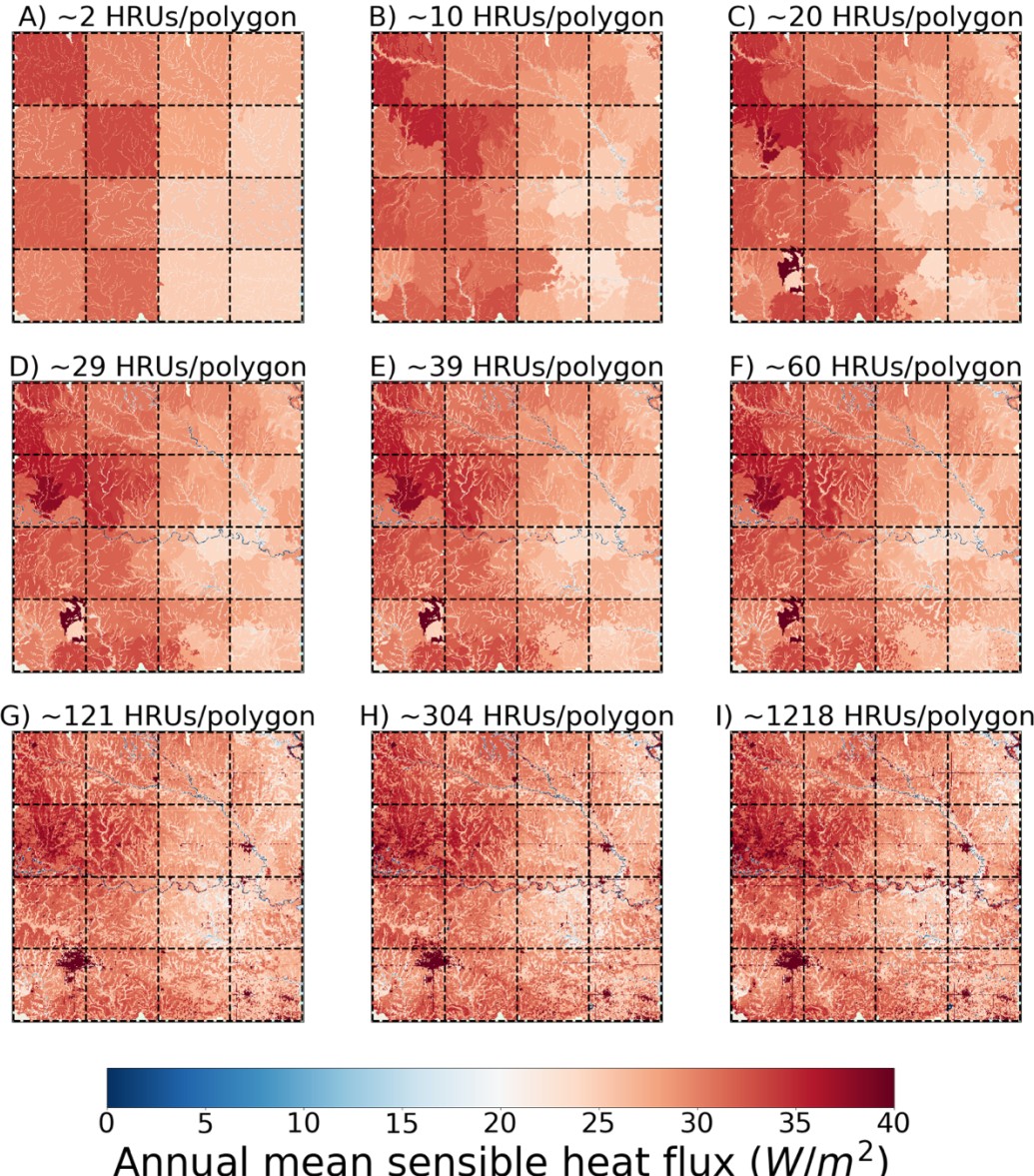

**Figure 9. Simulated annual mean sensible heat flux over the 1.0-degree SGP domain. Each panel shows the mean sensible heat flux from the different HydroBlocks simulations. Each simulation was run with different configurations of the revised hierarchical clustering algorithm. Simulations A-C increase $k$ from 1 to 5 to 10 while setting $p = 1$ and $n = 1000$; simulations D-F decrease $n$ from 5 to 3 to 2 while setting $p = 1$ and $k = 10$; simulations G-I increase the average number of intra-band clusters from 2 to 3 to 5 while setting $n = 2$ and $p = 5$. The average number of HRUs per macroscale polygon is shown as the title of each panel (the number can differ per macroscale polygon).**

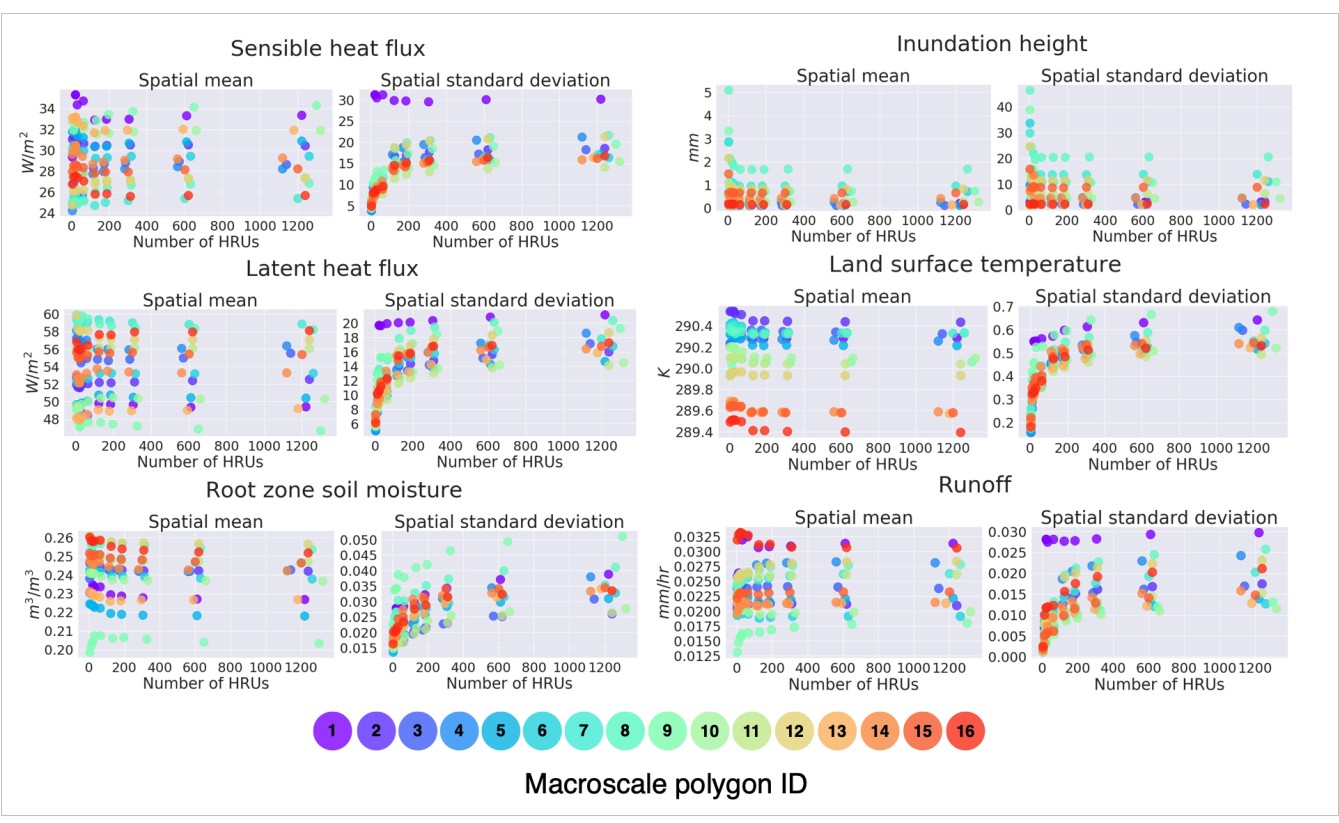

**Fig 10. Convergence of annual mean of the spatial mean and spatial standard deviation of runoff, land surface temperature, inundation height, root zone soil moisture, latent heat flux, and sensible heat flux for the 9 different HRU configurations. Each color denotes a different macroscale polygon in the 1.0-degree SGP domain.**

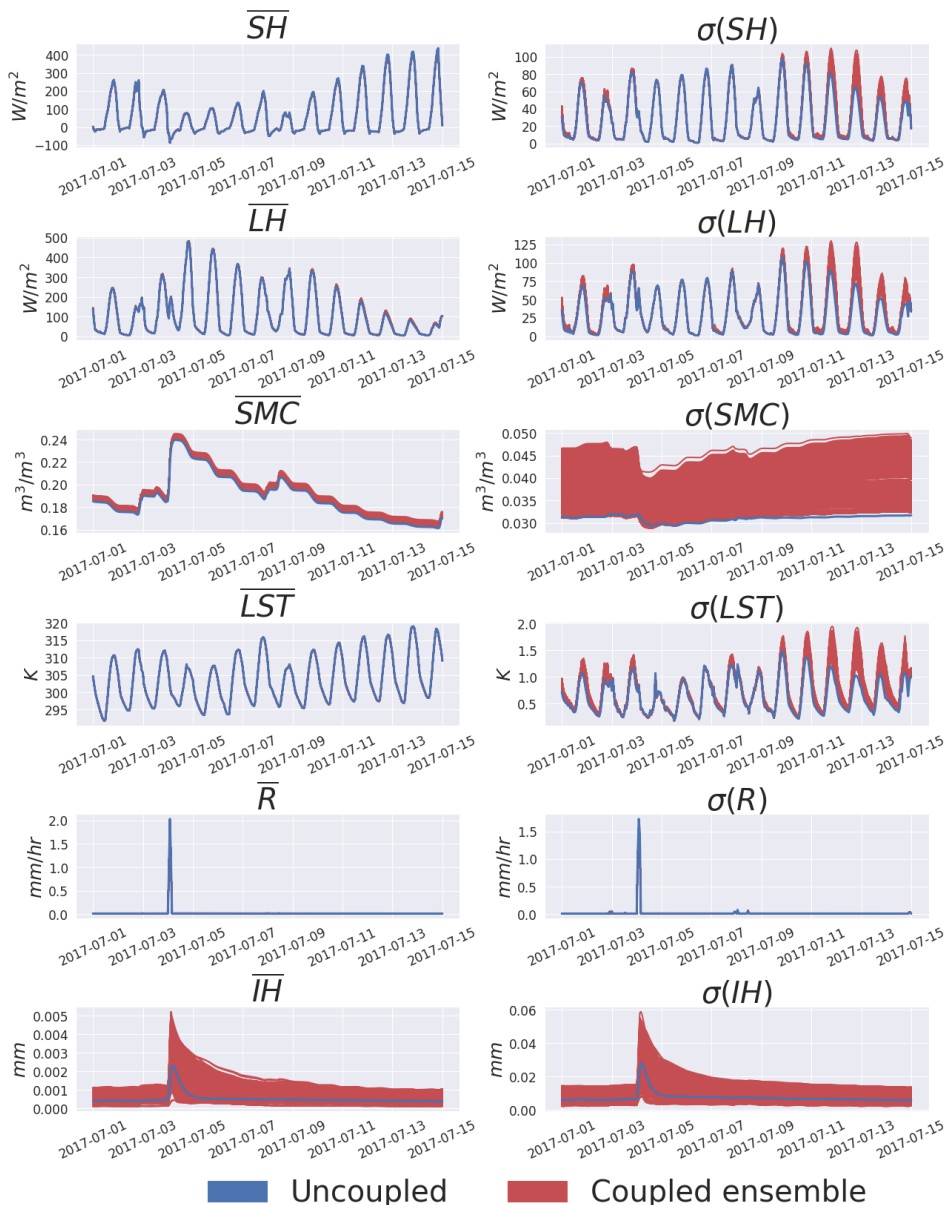

**Fig 11. Ensemble spread from the Sobol sensitivity analysis of the coupled routing/land surface HydroBlocks model between July 1st, 2017 and July 15th, 2017.** The shown simulated variables include the spatial mean ($\overline{X}$) and spatial standard deviation ($\sigma(X)$) of sensible heat flux ($SH$), latent heat flux ($LH$), root zone soil moisture ($SMC$), land surface temperature ($LST$), runoff ($R$), and inundation height ($IH$). For comparison, the baseline uncoupled simulation is shown in blue.

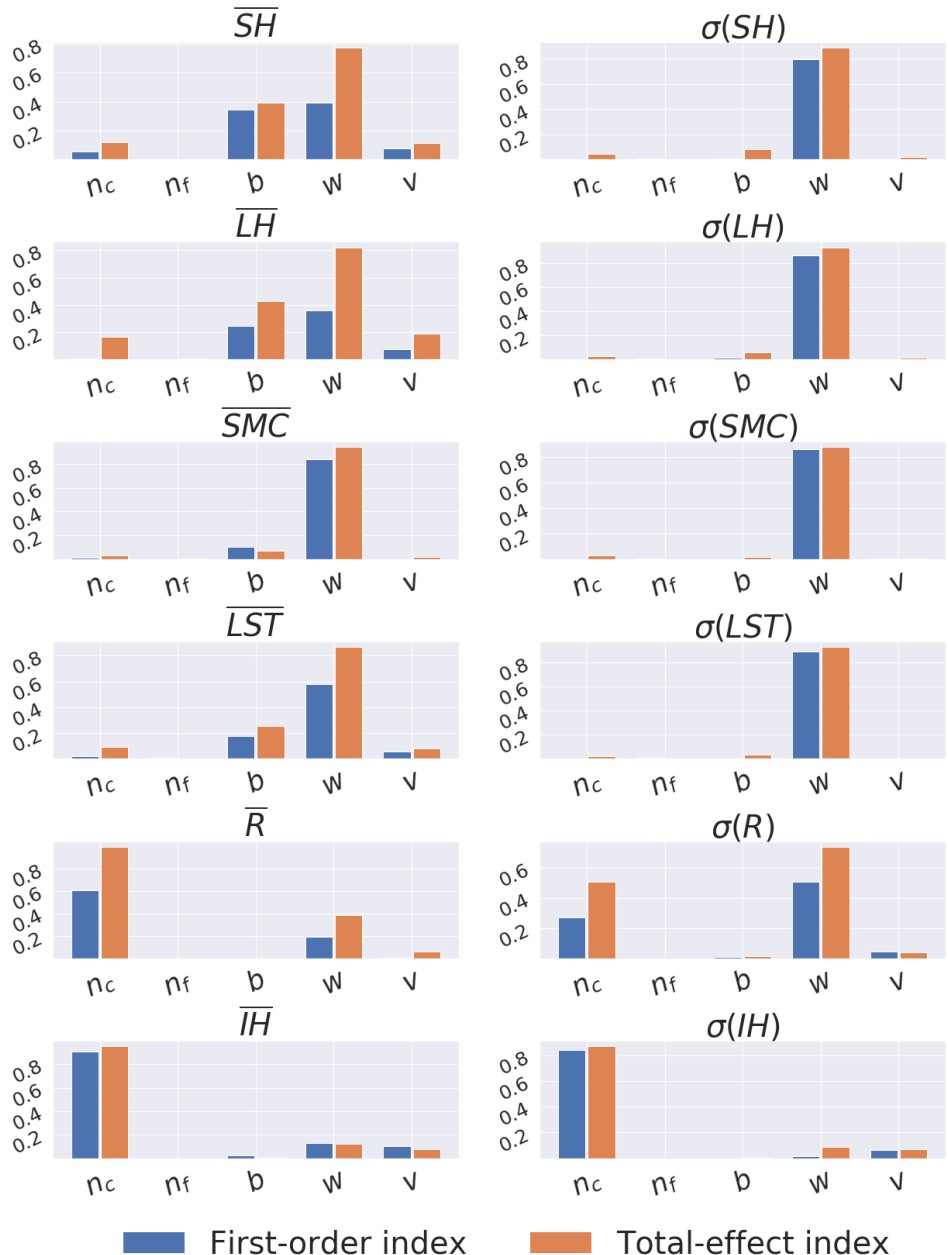

**Fig 12. First-order ($S_i$) and total-effect sensitivity ($S_{Ti}$) indices from the Sobol Sensitivity analysis of the $n_c$ (channel mannings roughness coefficient), $n_f$ (floodplain mannings roughness coefficient), $b$ (channel bankfull depth), $w$ (channel width), and $v$ (uniform overland flow velocity) parameters in the HydroBlocks land surface model over the SGP site in Oklahoma. The explored simulated variables are the same as those in Figure 11.**