# Peer review of "HydroBlocks v0.2: Enabling a field-scale two-way coupling between the land surface and river networks in Earth system models"

_Geoscientific Model Development, 2020_

## Short Comment (SC1) · 14 Nov 2020

Dear authors,

in my role as Executive editor of GMD, I would like to bring to your attention our Editorial version 1.2:

https://www.geosci-model-dev.net/12/2215/2019/

This highlights some requirements of papers published in GMD, which is also available on the GMD website in the 'Manuscript Types' section:

http://www.geoscientific-model-development.net/submission/manuscript_types.html

In particular, please note that for your paper, the following requirements have not been met in the Discussions paper:

- "The main paper must give the model name and version number (or other unique identifier) in the title."

- "If the model development relates to a single model then the model name and the version number must be included in the title of the paper. If the main intention of an article is to make a general (i.e. model independent) statement about the usefulness of a new development, but the usefulness is shown with the help of one specific model, the model name and version number must be stated in the title. The title could have a form such as, "Title outlining amazing generic advance: a case study with Model XXX (version Y)"."

In order to simplify reference to your developments, please add a model name (and/or its acronym) and a version number in the title of your article in your revised submission to GMD.

Yours,

Astrid Kerkweg

―――――――――――――――――――――

---

## Referee Comment (RC1) · Anonymous Referee #1 · 17 Nov 2020

This manuscript presents a high-resolution land-river coupling strategy in an earth system modeling context. The major conclusions are (I am directly quoting the authors): "1) the implementation of the two-way coupling between the land surface and the river network leads to appreciable differences in the simulated spatial heterogeneity of the surface energy balance; 2) a limited number of tiles ($\sim$300 per 0.25-degree cell) are required to approximate the fully distributed simulation adequately; 3) the surface energy balance partitioning is sensitive to the river routing model parameters." The study is properly motivated and overall well written. I do have a couple of major comments for the authors to consider.

[Figure]

1. The innovations could be better justified. It is intuitive that accounting for land-river two-way coupling will lead to non-negligible difference in the land surface water and energy balance, and high-resolution modeling of that will overall help to better capture spatial heterogeneity. This is not a very new understanding.

2. The benefits of this high-resolution land-river coupling strategy could be more clearly demonstrated. Typically, a new modeling strategy should help either reduce uncertainty or improve prediction. Uncertainty does not seem to be the focus here. Then how about improving prediction? Has it helped to improve the simulation of surface inundation, streamflow, or energy fluxes? In the study area, ARM SGP provides lots of observational data, but the authors did not show any comparison between the model simulations and observations.

3. The impulse response function at the HRU level is constructed in a simplified way, e.g., assuming uniform and constant velocity 0.1m/s. How would this simplification affect the model fidelity? Moreover, the impulse response function or unit hydrograph concept was originally developed at the small catchment scale, and theoretically it is not clear to me whether it can be applied at the HRU level. For instance, is the travel time histogram within a HRU statistically meaningful? Why not just use the kinematic wave routing method at the HRU level?
* * *

---

## Referee Comment (RC2) · Dai Yamazaki (Referee) · 18 Nov 2020

This manuscript represents the new two-way coupling scheme between land and river implemented in HydroBLOCK model. As the importance of surface water dynamics in land hydrology modelling and Earth system modelling is discussed recently, the model improvement proposed in this study has a contribution to the science community. The description of the model is mostly adequate, and the test simulation results look reasonable. I think the manuscript still need some improvement focusing on more detailed and kind description of the method, before acceptance.

Technical comments:

L139: "The basins are first delineated from a 30m DEM".

Please provide the definition of "basins". This is a specific technical concept in the developed model, and different from the general-use meaning. As far as I understand, the river network is divided into multiple "reaches", and the 30m pixels drained to each "reach" is defined as the "basin" corresponding to the reach. Also, I recommend to briefly explain how river channels and reaches are defined in this study. Even in the case this is mentioned in the previous paper (Chaney et al, 2016; 2018), the explanation will enhance the understanding of readers, as this is the core of the approach proposed in this manuscript.

L143: "These characteristic basins were identified using latitude, longitude, flow accumulation area, and the natural logarithm of the flow accumulations area as feature predictors."

Please explain the background reason of using these variables as input to clustering. (for example: log-scale accumulation area to separate the small hilltop basins from large rivers; lat-lon to represent the difference of atmospheric forcing by locations).

L150: "First, all channel grid cells within a given characteristic basin"

Please explain how the "channel grid cells" are defined. Also, it is better to provide some info on "what is grid cells, and what is macro-scale grids".

L152: "The binning involves creating groups of HAND values that have an areal coverage n (user-defined) times larger than its adjacent lower height band".

Please explain the background reason of this methodology? Why upstream band has larger area compared to downstream band?

L159: "to represent intra-band heterogeneity of land use, soils, and elevation, among others."

I think it is better to write the purpose of intra-band cluster implementation, rather than

explaining the parameters to define intra-band clusters. (i.e. representation of different land type is not the ultimate purpose, rather than that, I guess authors want to represent different land hydrological reactions due to the difference of land types, such as water and heat flux.).

L210: "much larger than many of the computed channel widths of the delineated streams (∼1 meter)" This assumption is only valid for small scale river basins. The authors should mention the limitation of this assumption, and further development is needed to apply the proposed method to large-scale rivers (for example, how river channel pixels are defined appropriately, if pixel size is smaller than river width? We do need additional data source and pre-processing in this case).

Section 2.3, Section 2.6, Figure 4:

The relationship among "reach", "basin", "characteristic basin", "height band", and "HRU" is not very clear, and I need to read this parts several times to understand the model structure. To improve the explanation, I suggest followings: - Update Figure 4D, or add another figure to explain the above relationship. Figure 4D is from the previous paper, and clustering approach of Figure 4D is not consistent to the explanation in this manuscript. I recommend to add a figure/panel to clearly explain the relationship between "characteristic basin, reach/basin, reach topography". - Clearly explain that "one characteristic basin has several reaches inside". "each reach has its corresponding basin, and each reach has height bands information to represent flood stage; these are used for the river routine component". (I suggest moving descriptions on delineation of the reach/basin topography just after Section 2.3, then readers can better understand the relationship between HRU generation and reach topography generation.

L222: "the inundation heights per height band are averaged across all basins that belong to a given characteristic basin (Figure 5B)"

By this process, the surface water extent in the lower bottom part of each height band is

distributed widely to the entire land surface of the corresponding height band, causing the overestimation of the inundated water surface. This will lead to the overestimation of the infiltration from floodplain to soil, and affect the heat and water flux accordingly. This should be discussed as the limitation of current approach. In addition, "Figure 5B" should be "Figure 4B".

L297: "lakes throughout the region."

Is it possible to explain how lakes are represented in the proposed model? As lakes are apparent in the result figures, some explanation should be essential.

L374: "the 16 interconnected cells take 5 minutes"

It is not clear what this "the 16 interconnected cells" corresponds to. Please clearly mention that this means "16 macroscale grids within the target 1deg domain. Also, it is better if authors mentioned the expected calculation cost for potential larger scale simulations, if HydroBLOCK is planned to be applied on continental or global scales.

L405: "One approach being explored by the co-authors is to cluster the lower stream orders."

This will also increase the discrepancy between "vector-shaped basins" and "rectangular macro-scale grid (and atmospheric forcing data as a result). This difficulty is also better to be mentioned.

L411: "update the boundary conditions iteratively"

It is not clear which "boundary conditions" authors want to mention in this sentence (e.g. upstream river inflow? Atmospheric forcing? Or between-basin horizontal water exchange?)

L424: "The flooding component of the scheme will then enable the valley to fill-up and, thus, producing a first-order representation of the time-varying reservoir spatial coverage."

This assumption is only valid for small-scale reservoirs which can be represented within a single grid. Further consideration is needed to represent large lakes/reservoirs which spans multiple grid boxes.
* * *

---

## Author Comment (AC1) · 21 Jan 2021

**Response to reviewers' comments**

"Two-way coupling between the sub-grid land surface and river networks in Earth system models" by N. W. Chaney, L. Torres-Rojas, N. Vergopolan, C. K. Fisher

We thank the reviewers for their time and helpful comments. We have addressed each point below. Reviewer comments are shown in *blue italics*, while author responses are shown in unformatted text.

**Executive director**: *Please note that for your paper, the following requirements have not been met in the Discussions paper: 1) "The main paper must give the model name and version number (or other unique identifier) in the title." 2) "If the model development relates to a single model then the model name and the version number must be included in the title of the paper. If the main intention of an article is to make a general (i.e. model independent) statement about the usefulness of a new development, but the usefulness is shown with the help of one specific model, the model name and version number must be stated in the title. The title could have a form such as, "Title outlining amazing generic advance: a case study with Model XXX (version Y)"." In order to simplify reference to your developments, please add a model name (and/or its acronym) and a version number in the title of your article in your revised submission to GMD.*

We thank the executive director for this feedback. We will add the model name and version number in the title of the revised manuscript.

**Reviewer #1**: *This manuscript presents a high-resolution land-river coupling strategy in an earth system modeling context. The major conclusions are (I am directly quoting the authors): "1) the implementation of the two-way coupling between the land surface and the river network leads to appreciable differences in the simulated spatial heterogeneity of the surface energy balance; 2) a limited number of tiles (~300 per 0.25-degree cell) are required to approximate the fully distributed simulation adequately; 3) the surface energy balance partitioning is sensitive to the river routing model parameters." The study is properly motivated and overall well written. I do have a couple of major comments for the authors to consider.*

We thank the reviewer for the constructive feedback. We provide responses to the reviewer's comments below.

*The innovations could be better justified. It is intuitive that accounting for land-river two-way coupling will lead to non-negligible difference in the land surface water and energy balance, and high-resolution modeling of that will overall help to better capture spatial heterogeneity. This is not a very new understanding.*

We agree that the role of a two-way coupling between the land surface and river network is not a new understanding and indeed it is known to play a large role in water limited regions that rely on recharge from upstream water sources (e.g., Nile river in Egypt). However, this process is almost completely missing in Earth system models where rivers mostly only receive water from the land surface but are unable to

recharge the surrounding regions. As such, the innovation of this work is to design a scheme that is able to effectively and efficiently model this process by enabling a two-way connection between the modeled rivers and the sub-grid tiling schemes.

*The benefits of this high-resolution land-river coupling strategy could be more clearly demonstrated. Typically, a new modeling strategy should help either reduce uncertainty or improve prediction. Uncertainty does not seem to be the focus here. Then how about improving prediction? Has it helped to improve the simulation of surface inundation, streamflow, or energy fluxes? In the study area, ARM SGP provides lots of observational data, but the authors did not show any comparison between the model simulations and observations.*

The main purpose of this paper is to develop a new parameterization that is able to couple the sub-grid approaches to river networks and land surface heterogeneity in Earth system models which remains a known weakness in these models. Although we certainly agree that the scheme should be evaluated with observations, an exhaustive evaluation of the scheme using observations is beyond the scope of this paper and is the focus of subsequent work among the co-authors. That being said, we agree that moving forward, there is a need to comprehensively evaluate the scheme and thus we will enhance the discussion section to note the need to evaluate the scheme with an emphasis on what available in-situ and remote sensing data could be used.

*The impulse response function at the HRU level is constructed in a simplified way, e.g., assuming uniform and constant velocity 0.1m/s. How would this simplification affect the model fidelity? Moreover, the impulse response function or unit hydrograph concept was originally developed at the small catchment scale, and theoretically it is not clear to me whether it can be applied at the HRU level. For instance, is the travel time histogram within a HRU statistically.*

We appreciate the reviewers feedback about the arbitrary and constraining impact of fixing the overland flow velocity to 0.1 m/s. As noted in the manuscript, the travel time histogram of the HRU to the reach is precalculated from the high-resolution DEM via path of steepest descent. In essence, the travel time of each 30-meter grid cell that belongs to a given HRU is calculated and then used to assemble the histogram of travel times. Since the fixed flow velocity assumes the flow to the channel is not impacted by more/less water down the hillslope, this assumption will be valid. We will clarify this assumption in the revised manuscript. It is also important to note that one could also use a kinematic wave on the hillslopes (height bands + HRUs) and this option will most likely be implemented in the near future (we will include this note in the revised manuscript). Time permitting we will also add a sensitivity analysis of how the uniform flow velocity impacts the modeling results.

**Reviewer #2**: *This manuscript represents the new two-way coupling scheme between land and river implemented in the HydroBlocks model. As the importance of surface water dynamics in land hydrology modelling and Earth system modelling is discussed recently, the model improvement proposed in this study has a contribution to the science community. The description of the model is mostly adequate, and the test*

*simulation results look reasonable. I think the manuscript still need some improvement focusing on more detailed and kind description of the method, before acceptance.*

We thank the reviewer for the constructive feedback. We provide responses to reviewers comments below.

*L139: "The basins are first delineated from a 30m DEM". Please provide the definition of "basins". This is a specific technical concept in the developed model, and different from the general-use meaning. As far as I understand, the river network is divided into multiple "reaches", and the 30m pixels drained to each "reach" is defined as the "basin" corresponding to the reach. Also, I recommend to briefly explain how river channels and reaches are defined in this study. Even in the case this is mentioned in the previous paper (Chaney et al, 2016; 2018), the explanation will enhance the understanding of readers, as this is the core of the approach proposed in this manuscript.*

Thank you for highlighting the importance to be more specific regarding our use of the term "basin". We will make its definition more explicit in the revised manuscript by focusing more on the term "reach". We will also provide further details on the algorithm used to delineate the rivers. Although the details are provided in Chaney et al., 2018, we agree that it would be useful to have it be more explicit in this paper as well.

*L143: "These characteristic basins were identified using latitude, longitude, flow accumulation area, and the natural logarithm of the flow accumulations area as feature predictors." Please explain the background reason of using these variables as input to clustering. (for example: log-scale accumulation area to separate the small hilltop basins from large rivers; lat-lon to represent the difference of atmospheric forcing by locations).*

We thank the reviewer for this great idea. As suggested, we will add more background for the use of each predictor in the clustering algorithm.

*L150: "First, all channel grid cells within a given characteristic basin". Please explain how the "channel grid cells" are defined. Also, it is better to provide some info on "what is grid cells, and what is macro-scale grids".*

The channel grid cells are computed from the flow accumulation area computed at 30 meters for the domain. The channel grid cells that belong to a characteristic basin are all the 30-meter grid cells defined as "channel" from the channel delineation algorithm that belong to a given clustered characteristic basin. We will clarify this in the revised manuscript as well as make a better distinction between the fine-scale and macro-scale grid cells.

*L152: "The binning involves creating groups of HAND values that have an areal coverage n (user-defined) times larger than its adjacent lower height band".*
*Please explain the background reason of this methodology? Why upstream band has larger area compared to downstream band?*

It is important to first note that the height bands will be upslope/downslope since these will be away from each channel and not how one normally thinks of upstream/downstream along the channel. With regards to the height band discretization, the purpose for this approach has to do mostly with the interest to "zoom" in on the region immediately surrounding each reach. Using the uniform discretization of height bands defined in Chaney et al., 2018 led to too coarse height bands in riparian areas; this led to too many height bands being necessary to ensure the floodplain dynamics "converged" (i.e., it didn't change that much the more height bands we added). With the added module, we changed the algorithm to have it have very high resolution around the reaches and then have the height bands become larger as we move away from the reach. Note that this does not mean that the final stage of intra-band clustering cannot still have a large number of HRUs in the upslope height bands; they just won't play as big of a role in the riparian dynamics so there is not a need to further increase the hillslope discretization. We will further clarify this in the revised manuscript.

*L159: "to represent intra-band heterogeneity of land use, soils, and elevation, among others." I think it is better to write the purpose of intra-band cluster implementation, rather than explaining the parameters to define intra-band clusters. (i.e. representation of different land type is not the ultimate purpose, rather than that, I guess authors want to represent different land hydrological reactions due to the difference of land types, such as water and heat flux.).*

This is an excellent suggestion, thank you. We agree that providing the reader with a reason why we are doing the intra-band clustering in the first place would help make the text more intuitive. We will add this clarification in the revised manuscript.

*L210: "much larger than many of the computed channel widths of the delineated streams (_1 meter)" This assumption is only valid for small scale river basins. The authors should mention the limitation of this assumption, and further development is needed to apply the proposed method to large-scale rivers (for example, how river channel pixels are defined appropriately, if pixel size is smaller than river width? We do need additional data source and pre-processing in this case).*

Although not mentioned currently in the text, the alternative described by the reviewer is already mostly implemented in the scheme as well. As in, if we know the channel width, we can have the channel HRU "grow" into the adjacent pixels to ensure the width matches the predicted/observed channel width. Given the minor effort required at this point to include this feature, we will add it to the model for the revised manuscript and mention this feature in the revision. In any case, this also brings up the important question of what data we are using to describe the channel width. Although for this paper we use regressions that were made for CONUS; we are fully aware that for continental-scale applications we would need to combine these regressions with the emerging global datasets of channel width and depth.

*Section 2.3, Section 2.6, Figure 4:*
*The relationship among "reach", "basin", "characteristic basin", "height band", and "HRU" is not very clear, and I need to read this parts several times to understand the*

*model structure. To improve the explanation, I suggest followings: - Update Figure 4D, or add another figure to explain the above relationship. Figure 4D is from the previous paper, and clustering approach of Figure 4D is not consistent to the explanation in this manuscript. I recommend to add a figure/panel to clearly explain the relationship between "characteristic basin, reach/basin, reach topography". – Clearly explain that "one characteristic basin has several reaches inside". "each reach has its corresponding basin, and each reach has height bands information to represent flood stage; these are used for the river routine component". (I suggest moving descriptions on delineation of the reach/basin topography just after Section 2.3, then readers can better understand the relationship between HRU generation and reach topography generation.*

We appreciate the reviewer's feedback. We understand how the current terminology can be confusing at times. We will revise the distinction between characteristic basin, reach/basin, and reach topography accordingly.

*L222: "the inundation heights per height band are averaged across all basins that belong to a given characteristic basin (Figure 5B)". By this process, the surface water extent in the lower bottom part of each height band is distributed widely to the entire land surface of the corresponding height band, causing the overestimation of the inundated water surface. This will lead to the overestimation of the infiltration from floodplain to soil, and affect the heat and water flux accordingly. This should be discussed as the limitation of current approach. In addition, "Figure 5B" should be "Figure 4B".*

As the reviewer suggests, this is one of the main limitations of the proposed method. However, we should note that this is a rationale for the clustering of the characteristic basins. Although, there will always be a limitation by grouping the floodplains of upstream/downstream reaches, the clustering ensures that although not perfect the two-way coupling can happen in lower order vs higher order streams, higher vs lower elevation, high vs low flow accumulation area, etc… In the end, this is the general rationale behind HydroBlocks, where it is a trade-off between fully representing the fully distributed simulation and ensuring computational tractability for implementation in large scale applications.

*L297: "lakes throughout the region." Is it possible to explain how lakes are represented in the proposed model? As lakes are apparent in the result figures, some explanation should be essential.*

As currently implemented, lakes are independent water bodies in the 1d land surface model that don't interact with the river network. We understand how the lakes can be confusing since it might seem that the lakes emerge from the routing model. While we do think that the proposed method could eventually represent them, it is misleading at this stage to make it seem like it is already represented. In the revised manuscript, we will make this distinction more explicit.

*L374: "the 16 interconnected cells take 5 minutes" It is not clear what this "the 16 interconnected cells" corresponds to. Please clearly mention that this means "16 macroscale grids within the target 1deg domain. Also, it is better if authors mentioned the expected calculation cost for potential larger scale simulations, if HydroBlocks is planned to be applied on continental or global scales.*

We will clarify this sentence in the revised manuscript. We will also discuss advantages and disadvantages of the scalability of the current algorithm. In the end, the full scalability won't be fully understood until it is run over the entire Contiguous United States (which is ongoing work); however, that work will be a follow-up to this paper and is thus seen as beyond the scope of this paper.

*L405: "One approach being explored by the co-authors is to cluster the lower stream orders." This will also increase the discrepancy between "vector-shaped basins" and "rectangular macro-scale grid (and atmospheric forcing data as a result). This difficulty is also better to be mentioned.*

The clustering of lower stream orders would only occur for networks that fall completely within each macroscale grid cell. Lower stream orders that cross grid cells would have to be resolved more explicitly. As the reviewer suggests, another option is to further adapt the grid cell to minimize cross-cell lower-stream orders. The future work will try these different concepts. In the end, the approach that minimizes computation and stays as close as possible to the regular grid will be adopted.

*L411: "update the boundary conditions iteratively" It is not clear which "boundary conditions" authors want to mention in this sentence (e.g. upstream river inflow? Atmospheric forcing? Or between-basin horizontal water exchange?)*

Given the implicit solver used, the upstream river inflow at each inlet reach in a given macroscale "grid cell" will need to be updated iteratively per time step (Picard iteration) to ensure convergence. Given that the study domain is a patchwork of different macroscale grid cells, this is necessary. We will clarify this section in the revised manuscript.

*L424: "The flooding component of the scheme will then enable the valley to fill-up and, thus, producing a first-order representation of the time-varying reservoir spatial coverage." This assumption is only valid for small-scale reservoirs which can be represented within a single grid. Further consideration is needed to represent large lakes/reservoirs which spans multiple grid boxes.*

Thank you for this feedback. We will add this clarification to the text. In any case, reservoirs would only be split from reservoirs covering multiple reaches since reaches are not split at the boundary In any case, we agree that reservoirs can (and will!) flood multiple reaches so cross-cell reservoirs will certainly exist; however, this should be able to be managed especially given the implemented iterations of the inflow/outflow boundary conditions. However, this will need to be tested in future work.

We would again like to thank the reviewers for their time and helpful comments.

---

## Author Response (AR1)

**Response to reviewers' comments**

"Two-way coupling between the sub-grid land surface and river networks in Earth system models" by N. W. Chaney, L. Torres-Rojas, N. Vergopolan, C. K. Fisher

We thank the reviewers for their time and helpful comments. We have addressed each point below. Reviewer comments are shown in *blue italics*, while author responses are shown in unformatted text.

**Executive director**: *Please note that for your paper, the following requirements have not been met in the Discussions paper: 1) "The main paper must give the model name and version number (or other unique identifier) in the title." 2) "If the model development relates to a single model then the model name and the version number must be included in the title of the paper. If the main intention of an article is to make a general (i.e. model independent) statement about the usefulness of a new development, but the usefulness is shown with the help of one specific model, the model name and version number must be stated in the title. The title could have a form such as, "Title outlining amazing generic advance: a case study with Model XXX (version Y)"." In order to simplify reference to your developments, please add a model name (and/or its acronym) and a version number in the title of your article in your revised submission to GMD.*

We thank the executive director for this feedback. We have added the model name and version number in the title of the revised manuscript.

**Reviewer #1**: *This manuscript presents a high-resolution land-river coupling strategy in an earth system modeling context. The major conclusions are (I am directly quoting the authors): "1) the implementation of the two-way coupling between the land surface and the river network leads to appreciable differences in the simulated spatial heterogeneity of the surface energy balance; 2) a limited number of tiles (~300 per 0.25-degree cell) are required to approximate the fully distributed simulation adequately; 3) the surface energy balance partitioning is sensitive to the river routing model parameters." The study is properly motivated and overall well written. I do have a couple of major comments for the authors to consider.*

We thank the reviewer for the constructive feedback. We provide responses to the reviewer's comments below.

*The innovations could be better justified. It is intuitive that accounting for land-river two-way coupling will lead to non-negligible difference in the land surface water and energy balance, and high-resolution modeling of that will overall help to better capture spatial heterogeneity. This is not a very new understanding.*

We agree that the role of a two-way coupling between the land surface and river network is not a new understanding and indeed it is known to play a large role in water limited regions that rely on recharge from upstream water sources (e.g., Nile river in Egypt). However, this process is almost completely missing in Earth system models where rivers mostly only receive water from the land surface but are unable to

recharge the surrounding regions. As such, the innovation of this work is to design a scheme that is able to effectively and efficiently model this process by enabling a two-way connection between the modeled rivers and the sub-grid tiling schemes. The penultimate paragraph in the introduction outlines the deficiencies in ESMs related to this weakness; the developed coupling approach would make it possible to address these issues.

*The benefits of this high-resolution land-river coupling strategy could be more clearly demonstrated. Typically, a new modeling strategy should help either reduce uncertainty or improve prediction. Uncertainty does not seem to be the focus here. Then how about improving prediction? Has it helped to improve the simulation of surface inundation, streamflow, or energy fluxes? In the study area, ARM SGP provides lots of observational data, but the authors did not show any comparison between the model simulations and observations.*

The main purpose of this paper is to develop a new parameterization that is able to couple the sub-grid approaches to river networks and land surface heterogeneity in Earth system models which remains a known weakness in these models. Although we certainly agree that the scheme should be evaluated with observations, an exhaustive evaluation of the scheme using observations is beyond the scope of this paper and is the focus of subsequent work among the co-authors.

*The impulse response function at the HRU level is constructed in a simplified way, e.g., assuming uniform and constant velocity 0.1m/s. How would this simplification affect the model fidelity? Moreover, the impulse response function or unit hydrograph concept was originally developed at the small catchment scale, and theoretically it is not clear to me whether it can be applied at the HRU level. For instance, is the travel time histogram within a HRU statistically.*

We appreciate the reviewer's feedback about the arbitrary and constraining impact of fixing the overland flow velocity to 0.1 m/s. As noted in the manuscript, the travel time histogram of the HRU to the reach is precalculated from the high-resolution DEM via path of steepest descent. In essence, the travel time of each 30-meter grid cell that belongs to a given HRU is calculated and then used to assemble the histogram of travel times. Since the fixed flow velocity assumes the flow to the channel is not impacted by more/less water down the hillslope, this assumption will be valid. It is also important to note that one could also use a kinematic wave on the hillslopes (height bands + HRUs) and this option will most likely be implemented in the near future (we included this note in the revised manuscript). Section 2.4 of the revised manuscript mentions how the 0.1 m/s choice is arbitrary and can be set differently per HRU by the model user.

**Reviewer #2**: *This manuscript represents the new two-way coupling scheme between land and river implemented in the HydroBlocks model. As the importance of surface water dynamics in land hydrology modelling and Earth system modelling is discussed recently, the model improvement proposed in this study has a contribution to the science community. The description of the model is mostly adequate, and the test*

*simulation results look reasonable. I think the manuscript still need some improvement focusing on more detailed and kind description of the method, before acceptance.*

We thank the reviewer for the constructive feedback. We provide responses to the reviewer's comments below.

*L139: "The basins are first delineated from a 30m DEM". Please provide the definition of "basins". This is a specific technical concept in the developed model, and different from the general-use meaning. As far as I understand, the river network is divided into multiple "reaches", and the 30m pixels drained to each "reach" is defined as the "basin" corresponding to the reach. Also, I recommend to briefly explain how river channels and reaches are defined in this study. Even in the case this is mentioned in the previous paper (Chaney et al, 2016; 2018), the explanation will enhance the understanding of readers, as this is the core of the approach proposed in this manuscript.*

Thank you for highlighting the importance of being more specific regarding our use of the term "basin". In the updated manuscript we have replaced the term "basin" with "watershed" for ease of understanding. We also provide more details on how the reaches and watersheds are delineated in Section 2.3. Finally, we make a more explicit connection between reach and watershed by adding the following text "The watersheds are then assembled by finding all 30-meter pixels that flow into a given reach via steepest descent" in Section 2.3.

*L143: "These characteristic basins were identified using latitude, longitude, flow accumulation area, and the natural logarithm of the flow accumulations area as feature predictors." Please explain the background reason of using these variables as input to clustering. (for example: log-scale accumulation area to separate the small hilltop basins from large rivers; lat-lon to represent the difference of atmospheric forcing by locations).*

We thank the reviewer for this great idea. As suggested, we added more background for the use of each predictor in the clustering algorithm in Section 2.3.

*L150: "First, all channel grid cells within a given characteristic basin". Please explain how the "channel grid cells" are defined. Also, it is better to provide some info on "what is grid cells, and what is macro-scale grids".*

The channel grid cells are computed from the flow accumulation area computed at 30 meters for the domain. The channel grid cells that belong to a characteristic basin (now defined as cluster of watersheds in the updated manuscript) are all the 30-meter grid cells defined as "channel" from the channel delineation algorithm that belong to a given cluster of watersheds. To facilitate readability, the revised manuscript now specifies 30-meter pixel/fine-scale pixel. Furthermore, we no longer use the term grid cell or macroscale grid cell, because the updated manuscript uses the term macroscale polygon; this reflects the fact that the domain composition does not need to follow a regular grid.

*L152: "The binning involves creating groups of HAND values that have an areal coverage n (user-defined) times larger than its adjacent lower height band".*
*Please explain the background reason of this methodology? Why upstream band has larger area compared to downstream band?*

It is important to first note that the height bands will be upslope/downslope since these will be away from each channel and not how one normally thinks of upstream/downstream along the channel. With regards to the height band discretization, the purpose for this approach has to do mostly with the interest to "zoom" in on the region and add a more detailed characterization of the area immediately surrounding each reach. Using the uniform discretization of height bands defined in Chaney et al., 2018 led to too coarse height bands in riparian areas; this led to too many height bands being necessary to ensure the floodplain dynamics "converged". With the added module, we changed the algorithm to have it have a very high resolution around the reaches and then have the height bands become larger as we move away from the reach. Note that this does not mean that the final stage of intra-band clustering cannot still have a large number of HRUs in the upslope height bands; they just won't play as big of a role in the riparian dynamics so there is not a need to further increase the hillslope discretization. We have clarified this in the revised manuscript in Section 2.3.

*L159: "to represent intra-band heterogeneity of land use, soils, and elevation, among others." I think it is better to write the purpose of intra-band cluster implementation, rather than explaining the parameters to define intra-band clusters. (i.e. representation of different land type is not the ultimate purpose, rather than that, I guess authors want to represent different land hydrological reactions due to the difference of land types, such as water and heat flux.).*

This is an excellent suggestion, thank you. We agree that providing the reader with a reason why we are doing the intra-band clustering in the first place would help make the text more intuitive. We have added this clarification in the revised manuscript.

*L210: "much larger than many of the computed channel widths of the delineated streams (_1 meter)" This assumption is only valid for small scale river basins. The authors should mention the limitation of this assumption, and further development is needed to apply the proposed method to large-scale rivers (for example, how river channel pixels are defined appropriately, if pixel size is smaller than river width? We do need additional data source and pre-processing in this case).*

Section 4.4 of the revised manuscript addresses the fact that existing vector river network databases should be used to define the modeled river networks moving forward. This section also acknowledges the weakness of the current scheme in that it can only handle river widths that are at most the size of the fine-scale pixels from which the channels were delineated. This does not need to be the case, and will be revised in subsequent work on this coupling approach.

*Section 2.3, Section 2.6, Figure 4:*

*The relationship among "reach", "basin", "characteristic basin", "height band", and "HRU" is not very clear, and I need to read this parts several times to understand the model structure. To improve the explanation, I suggest followings: - Update Figure 4D, or add another figure to explain the above relationship. Figure 4D is from the previous paper, and clustering approach of Figure 4D is not consistent to the explanation in this manuscript. I recommend to add a figure/panel to clearly explain the relationship between "characteristic basin, reach/basin, reach topography". – Clearly explain that "one characteristic basin has several reaches inside". "each reach has its corresponding basin, and each reach has height bands information to represent flood stage; these are used for the river routine component". (I suggest moving descriptions on delineation of the reach/basin topography just after Section 2.3, then readers can better understand the relationship between HRU generation and reach topography generation.*

We appreciate the reviewer's feedback. We agree that the original terminology was confusing. The revised manuscript now uses the term watershed instead of basin. We have also moved away from the concept of characteristic basin; instead we use the term "cluster of watersheds" which is much more reflective of what is actually going on. The concept of characteristic/representative watershed can then be drawn from a given cluster of watersheds. We have also added a paragraph at the end of section 2.3 that summarizes how each sub-polygon is divided; here we explicitly mention the distinction between watershed and characteristic watershed. We also make clear that the routing module works on each "real" (i.e., non-clustered) reach while the land surface model works with the clusters of watersheds. We have updated the figures to improve comprehension.

*L222: "the inundation heights per height band are averaged across all basins that belong to a given characteristic basin (Figure 5B)". By this process, the surface water extent in the lower bottom part of each height band is distributed widely to the entire land surface of the corresponding height band, causing the overestimation of the inundated water surface. This will lead to the overestimation of the infiltration from floodplain to soil, and affect the heat and water flux accordingly. This should be discussed as the limitation of current approach. In addition, "Figure 5B" should be "Figure 4B".*

As the reviewer suggests, this is one of the main limitations of the proposed method. However, we should note that this is a rationale for the clustering of watersheds. Although, there will always be a limitation by grouping the floodplains of upstream/downstream reaches, the clustering ensures that although not perfect, the two-way coupling can happen in lower order vs higher order streams, higher vs lower elevation, high vs low flow accumulation area, etc… In the end, this is the general rationale behind HydroBlocks, it is a trade-off between fully representing the fully distributed simulation and ensuring computational tractability for implementation in large-scale applications. The revised manuscript now includes a discussion on this limitation in Section 4.1 and presents a possible path forward.

*L297: "lakes throughout the region." Is it possible to explain how lakes are represented in the proposed model? As lakes are apparent in the result figures, some explanation should be essential.*

As currently implemented, lakes are independent water bodies in the 1d land surface model that don't interact with the river network. We understand how the lakes can be confusing since it might seem that the lakes emerge from the routing model. While we do think that the proposed method could eventually represent them, it is misleading at this stage to make it seem like it is already represented. In the revised manuscript, we make this distinction more explicit and enhance the discussion to provide a path forward to formally include lakes and reservoirs in the coupled scheme.

*L374: "the 16 interconnected cells take 5 minutes" It is not clear what this "the 16 interconnected cells" corresponds to. Please clearly mention that this means "16 macroscale grids within the target 1deg domain. Also, it is better if authors mentioned the expected calculation cost for potential larger scale simulations, if HydroBlocks is planned to be applied on continental or global scales.*

We have clarified this sentence in the revised manuscript. We also discuss advantages and disadvantages of the scalability of the current algorithm in Sections 4.5 and 4.6. In the end, the full scalability of the routing scheme won't be fully understood until it is run over the entire Contiguous United States (which is ongoing work); however, that work will be a follow-up to this paper and is thus seen as beyond the scope of this paper. The discussion now mentions the need to further investigate the computational scaling properties of the algorithm.

*L405: "One approach being explored by the co-authors is to cluster the lower stream orders." This will also increase the discrepancy between "vector-shaped basins" and "rectangular macro-scale grid (and atmospheric forcing data as a result). This difficulty is also better to be mentioned.*

The clustering of lower stream orders would only occur for networks that fall completely within each macroscale polygon. Lower stream orders that cross polygons would have to be resolved more explicitly. As the reviewer suggests, another option is to further adapt the polygon to minimize cross-cell lower-stream orders. The future work will try these different concepts. In the end, the approach that minimizes computation and stays as close as possible to the regular grid will be adopted. We have amended the text to clarify where the clustering of the stream orders would occur.

*L411: "update the boundary conditions iteratively" It is not clear which "boundary conditions" authors want to mention in this sentence (e.g. upstream river inflow? Atmospheric forcing? Or between-basin horizontal water exchange?)*

Given the implicit solver used, the upstream river inflow at each inlet reach in a given macroscale polygon will need to be updated iteratively per time step (Picard iteration) to ensure convergence. Given that the study domain is a patchwork of different macroscale polygons, this is necessary. In any case, after careful consideration, we think

that these details are unnecessary in the manuscript. We have removed this paragraph from the revised manuscript to avoid confusion.

*L424: "The flooding component of the scheme will then enable the valley to fill-up and, thus, producing a first-order representation of the time-varying reservoir spatial coverage." This assumption is only valid for small-scale reservoirs which can be represented within a single grid. Further consideration is needed to represent large lakes/reservoirs which spans multiple grid boxes.*

Thank you for this feedback. We have added this clarification to the text. In any case, reservoirs would only be split from reservoirs covering multiple reaches since reaches are not split at the boundary In any case, we agree that reservoirs can (and will!) flood multiple reaches so cross-cell reservoirs will certainly exist; however, this should be able to be managed especially given the implemented iterations of the inflow/outflow boundary conditions. However, this will need to be tested in future work.

We would again like to thank the reviewers for their time and helpful comments.

---

## Author Response (AR2)

**Response to reviewers' comments**

"Two-way coupling between the sub-grid land surface and river networks in Earth system models" by N. W. Chaney, L. Torres-Rojas, N. Vergopolan, C. K. Fisher

We thank the reviewers for their time and helpful comments. We have addressed each point below. Reviewer comments are shown in *blue italics*, while author responses are shown in unformatted text.

**Reviewer #1**: *I appreciate the authors' responses, which addressed part of my comments. I still would like to challenge the authors to further elevate this manuscript. In particular, I have the following feedback on the authors' responses.*

We thank the reviewer for the constructive feedback. We provide responses to the reviewer's comments below.

*It is true that most ESMs don't represent two-way coupling between their land and river components. A complete representation of it involves both the suitable process description and, more importantly, a proper parameterization strategy. The former is relatively easy since the ESM modelers just need to borrow some governing equations from the well-established fields of hydrology or hydraulics. The latter, however, is much more challenging and, in my opinion, more critical in a model development paper like this. For a new process description in ESMs, a good parameterization usually means the parameter values are provided via some a priori estimation or extensive calibrations and ready to use for the potential users. As we are all aware, for ESMs it is usually not practical to expect the users to define/calibrate the parameter values by themselves because 1) it is computationally prohibitive to perform sufficient parameter calibration with ESMs like we used to do with much cheaper hydrological models and 2) very often the ESMs users have a very diversified background and many of them do not have adequate hydrology background to carry out such parameter estimation on their own. In a nutshell, when adding a new process representation in ESMs, it is most important to demonstrate that its corresponding parameterization strategy is compatible with the process description and hence effectively improves the model predictions in some aspects. In some sense, without a proper parameterization, adding a new process description will not necessarily bring better model predictions, particularly for sophisticated models like ESMs. Therefore, it is not obviously beyond the scope of this study to add sufficient model evaluations against the ARM SGP observations that the authors already have, and equally importantly, other observations at a regional or global scale where ESMs are typically applied at. In any case, it is my understanding that, when publishing a new process development in ESMs, it is more important to have a good parameterization strategy and adequately validate the new development against some observations. If the editors feel that it is not necessarily the standard for GMD, I am also fine with it.*

We appreciate the honest feedback from the reviewer. We agree that a parameterization aimed for use within ESMs should show improvement when compared to observations. However, determining the observations one uses to evaluate a parameterization like this one is not as trivial. Let's take for example the

spatial means of surface fluxes and land surface temperature over the entire domain and compare them to the VARANAL database which provides macroscale observations of surface fluxes derived from network of eddy covariance stations and energy balance/bowen ratio in-situ stations overvthe SGP domain. The figure below compares the modeled average summer of 2017 diurnal cycle of surface latent heat flux, sensible heat flux, and skin temperature of the coupled and uncoupled (baseline) simulations against the observations.

[Figure]

The coupling leads to a minor increase in macroscale latent heat flux and a minor decrease in sensible heat flux. When compared to the observations this leads to a slight (but almost negligible) improvement in SH and a slight deterioration in LH. Based on this one could conclude using the baseline model parameters that the implemented parameterization doesn't improve the macroscale spatial mean (which is already clear in Figure 8). A comparison with discharge would most likely show almost the exact same story. However, we argue that that conclusion would be misguided as there are large changes in the modeled spatial heterogeneity of the system which would play a key role in the land-atmopshere interactions over the system. Furthermore, the implemented parameterization enables the model to represent key land/river interface processes that are known to be important yet are almost completely lacking in ESMs. As such, we argue that simply evaluating and calibrating the parameterizations against the avalaible macroscale spatial mean observations at SGP is not appropriate (and strongly misleading).

Instead, it should be evaluated using observed remote sensing spatial fields of surface fluxes, inundation, and land surface temperature. One might imagine using MODIS LST or Ecostress LST to perform this evaluation/calibration. And indeed the co-authors have an ongoing study in which they are doing just that. However, we strongly argue that that type of evaluation is out of the scope of this study as it involves multiple datasets, a new calibration strategy for tiling schemes, etc… As such we aim to publish that work in a different study. Furthermore, simply including the comparison to the spatial mean VARANAL data would be highly misleading at it misses the point as to what we are aiming to accomplish with this work. For these reasons, we are leaving the evaluation/calibration for a subsequent paper and instead focus exclusively here on the description of the new parameterization and the sensitivity analysis to explore the parameterization. That being said, we agree that the paper should mention the planned strategy to compare to observations. We have updated the discussion to describe a path towards properly validating and calibrating the parameterization; this is now discussed in the first subsection of the discussion section.

Moreover, the explanation of using a constant and uniform velocity for the HRU-level impulse response function is not very convincing. The authors stated that "Since the fixed flow velocity assumes the flow to the channel is not impacted by more/less water down the hillslope, this assumption will be valid." This assumption "the flow to the channel is not impacted by more/less water down the hillslope" is not quite rigorous. At an hourly or shorter time step, this assumption is certainly not correct, because overland flow velocity and the travel time will be affected by the surface runoff depth, as indicated by the well-known kinematic wave equations, e.g., Manning's equation. At a daily time step, such a statement may be ok, if slightly rephrased, since the impacts of runoff depth are somehow negligible in terms of travel time, but then all the other advantages of having two-way land-river interactions and sub-grid heterogeneities might not be significant at a daily or longer time step as well.

We thank the reviewer for this constructive feedback. We agree that implementing a kinematic wave to solve surface runoff along the hillslopes would be a valuable contribution moving forward. As the reviewer points out, there will be important limitations when it comes to modeling the flow velocity when we are considering ponding in regions near the channel. To evaluate its impact we have enhanced the sensitivity analysis performed in this study by including the uniform flow velocity parameter. We also now use a Sobol sensitivity analysis—a more formal variance-based sensitivity analysis to evaluate the role of each parameter. As shown in Figure 12, the results show that although the model is not as sensitive to the uniform overland flow velocity parameters as other channel-specific parameters, its impact should not be ignored. As such, in future updates to HydroBlocks we will add a kinematic wave solver for overland flow along the hillslopes. However, for this paper, we are comfortable leaving the uniform velocity approach. We have updated the discussion to indicate the need think more critically about this challenge and how it could be addresed moving forward.

We would again like to thank the reviewer for their time and helpful comments.

---

## Author Response (AR3)

**Response to reviewers' comments**

"Two-way coupling between the sub-grid land surface and river networks in Earth system models" by N. W. Chaney, L. Torres-Rojas, N. Vergopolan, C. K. Fisher

We thank the reviewers for their time and helpful comments. We have addressed each point below. Reviewer comments are shown in *blue italics*, while author responses are shown in unformatted text.

**Associate Editor #1**: *The discussion between the reviewer and yourself has convinced me that this is really half a model description with the parametrisation approach to come later. Given the complexity of the model I think a standalone description is worth publishing and that GMD is the appropriate journal for that. However, since the relative performance of the new model is not well understood this should be made very clear to the reader. This is done at the start of the discussion "it remains unclear if this paramerterization improves the macroscale modelling of surface fluxes and inundation…" But I think a similar statement should be made either in the aims, conclusions, or both. Such that it's clear that further research is necessary before the model description is complete.*

We appreciate the topical editor for his evaluation of the paper and feedback. We would argue that the implemented two-way coupling between sub-grid tiles and river routing schemes is a parameterization as it enables a way to include this process in a simple yet efficient manner in Earth system models. That being said, we agree that the parameterization has not been robustly evaluated against observations at this point and thus requires further evaluation. As a result, as suggested, we have added the following sentences at the end of the conclusion section:

*"Finally, the implemented scheme shows an appreciable impact on the modeled spatial heterogeneity (e.g., spatial variance) of surface fluxes and states; however, its influence on the macroscale spatial means is relatively small. In other words, although the two-way coupling parameterization offers potential to provide more realistic simulated multi-scale spatial patterns, its impact on improving the domain-average response appears to be limited and requires further evaluation."*